# Multicomponent and multisensory communicative acts in orang-utans may serve different functions

Marlen Fröhlich [1✉], Natasha Bartolotta[1], Caroline Fryns[1], Colin Wagner[2], Laurene Momon [2], Marvin Jaffrezic[2], Tatang Mitra Setia[3], Maria A. van Noordwijk[1] & Carel P. van Schaik[1,4]

From early infancy, human face-to-face communication is multimodal, comprising a plethora of interlinked communicative and sensory modalities. Although there is also growing evidence for this in nonhuman primates, previous research rarely disentangled production from perception of signals. Consequently, the functions of integrating articulators (i.e. production organs involved in multicomponent acts) and sensory channels (i.e. modalities involved in multisensory acts) remain poorly understood. Here, we studied close-range social interactions within and beyond mother-infant pairs of Bornean and Sumatran orang-utans living in wild and captive settings, to examine use of and responses to multicomponent and multisensory communication. From the perspective of production, results showed that multicomponent acts were used more than the respective unicomponent acts when the presumed goal did not match the dominant outcome for a specific communicative act, and were more common among non-mother-infant dyads and Sumatran orang-utans. From the perception perspective, we found that multisensory acts were more effective than the respective unisensory acts, and were used more in wild compared to captive populations. We argue that multisensory acts primarily facilitate effectiveness, whereas multicomponent acts become relevant when interaction outcomes are less predictable. These different functions underscore the importance of distinguishing between production and perception in studies of communication.

[1] Department of Anthropology, University of Zurich, Zurich, Switzerland. [2] DEPE-IPHC – Département Ecologie, Physiologie et Ethologie, University of Strasbourg, Strasbourg, France. [3] Fakultas Biologi, Universitas Nasional, Jakarta Selatan, Indonesia. [4] Center for the Interdisciplinary Study of Language Evolution (ISLE), University of Zurich, Zurich, Switzerland. ✉email: marlen.froehlich@uzh.ch

Human face-to-face communication is a multimodal phenomenon: our everyday speech is embedded in an interactional exchange of coordinated visual, auditory, and often even tactile signals. Some parts of these complex displays are intrinsically coupled due to the effort of vocal production (such as mouth movement accompanying speech sounds), but others are flexible (e.g. gaze and co-speech gestures). Research on the nature and function of human multimodal interaction has focused particularly on flexible combinations of different articulators (i.e. signal production organs such as hands, lips and eyes)[1,2]. For instance, speech acts accompanied by gestures and gaze are processed faster[3] and elicit faster responses[4,5], respectively. This suggests that a complex orchestration of articulators and sensory channels facilitates comprehension and prediction during language processing[6]. Many non-human species also have a natural predisposition for multimodal social interactions, as evident in complex mating, warning and dominance displays[7,8].

Multimodal signalling can be disentangled based on the perspective of production versus perception: multicomponent (or: multiplex) communication involves at least two different articulators or communication organs at the production side[6], such as hands plus gaze, whereas multisensory (or: multimodal sensu stricto) communication involves at least two different sensory channels at the perception end, such as visual plus auditory[9]. Many communicative acts are both multi-component and multisensory, for instance, a tactile gesture combined with a vocalisation, whereas some are just multisensory, such as the audio-visual loud scratch gesture (observed during the initiation of mother-offspring joint travel in chimpanzees and orangutans[10,11]), and others are only multicomponent, such as a visual gesture combined with a facial expression. In fact, our closest living relatives, the great apes, are renowned for signalling intentional communicative acts in large part by non-vocal means in their close-range dyadic interactions[8,12,13]. Many of these signals are intrinsically multisensory (e.g. tactile gestures that can be simultaneously seen and felt by a receiver, or lip-smacking which can be seen and heard), but they can also be integrated with other, non-vocal or vocal acts in multicomponent signal combinations (e.g.[12–14]). The term multimodality has confusingly been used to refer to communicative acts that involve multiple communicative features/articulators (e.g.[15,16]), but also multiple sensory channels (e.g.[7,17]). Therefore, we will henceforth refer to multicomponent and multisensory acts, respectively, to explicitly discriminate between the aspects of communicative acts that reflect production and affect perception, respectively (Table 1).

The fact that close-range communicative acts may be either multicomponent or multisensory (even if many are both) highlights the importance of teasing apart production and perception aspects of communicative acts if we wish to assess whether they serve different communicative functions. Studying the flexible production of signals is critical as some communication systems (e.g. those of primate species) often lack the one-to-one correspondence between signal and outcome[8,17]. On the other hand, understanding the role of perception is important because the function of animal signals is predicted by receiver psychology[18,19] and thus by the receiver's sensory systems[7,20]. However, to date, no study has explicitly examined specifically how the function of multicomponent signals compares to that of multisensory signals in a great ape taxon (nor, to our knowledge, in humans). The theoretical and empirical differences between these combination types are often ignored in comparative research[12,17], but addressing them would be key to draw conclusions about homologous features in the human/ape communication system[21].

A neurobiological perspective underscores the plausibility of differentiating between the production and perception aspects of communicative acts: in contexts or situations requiring a multicomponent act, the signaller is forced to execute (at least) two different motor commands in different articulators. Neurobiological research on human communicative processing suggests that the integration between speech and gesture depends on the context and is under voluntary control rather than obligatory[22]. Co-speech gestures may therefore provide additional information, depending on the communicative nature of the situation (e.g. whether or not there is shared common ground between the signaller and the recipient)[23] as well as on gaze direction (i.e. whether or not the signaller's gaze is directed at the addressee)[24]. Together with rich evidence that multicomponent acts serve to refine messages[1,25,26], this suggests they are of particular relevance when outcomes are less predictable: when social partners are less familiar or more socially distant to each other, they are less likely to have engaged in a specific communicative context, and disambiguation of signal meaning may become necessary.

The multisensory case explicitly takes the recipient's (and thus, the perception) perspective: the recipient is forced to integrate incoming information in at least two different sensory channels that initially are processed in different brain regions. Visual and auditory pathways, for instance, are largely separate before converging in the ventrolateral prefrontal cortex (vlPFC) onto neurons that represent higher-order multisensory representations of signals, such as vocalisations and their associated facial expressions[27]. This need to integrate may make it more likely that the communicative act is accurately processed compared to a unisensory signal, suggesting that multisensory communication serves to ensure that a signal is understood[28,29].

These neurobiological considerations suggest that multicomponent and multisensory acts may serve different functions. Comparative researchers have recently begun to study the function of great apes' multicomponent and multisensory communication via observational research, focusing on bi-articulatory gesture-vocal combinations[12–14,16] and considering mostly two

**Table 1 Definition and operationalisation of relevant terms used in analyses.**

| Term | Definition | Coding for GLMMs |
|---|---|---|
| Communicative act (CA) | Socially directed, mechanically ineffective movements of the face, head, limbs, or body, or vocalisations | Rows of dataset |
| Articulator | Organ involved in the production of CAs: manual, bodily, gaze, facial and vocal | Each 2 levels, e.g. not manual (0), manual (1) |
| Sensory channel | Modality involved in the perception of CAs: visual, tactile, auditory, seismic | Each 2 levels, e.g. not visual (0), visual (1) |
| Multisensory acts | CAs perceived through at least two different sensory channels | 2 Levels: unisensory (0), multisensory (1) |
| Multi-component acts | CAs involving at least two different articulators | 2 Levels: uni-component (0), multicomponent (1) |
| Effectiveness | Presence of apparently satisfactory outcome (ASO) sensu[11] | 2 Levels: other than ASO (0), ASO (1) |
| Dominant outcome match | Presumed goal sensu[63] matches interaction outcome that is most commonly (>50%) attributed to a specific CA sensu[75]. | 2 Levels: non-dominant (0), dominant (1) |

different function(s): redundancy and refinement[9,17,29] (but see e.g. refs. [17,30] for further hypotheses that have been discussed in relation to complex signal function). The redundant signal (hereafter referred to as redundancy) hypothesis implies that the different components convey the same information[9], facilitating the detection and processing of a message[28]. For example, using a conspicuous signal involving multiple modalities that contain the same information (e.g. audible and visual) makes it easier to be detected by a recipient in noisy environments[31,32] and can thus increase effectiveness (i.e. responsiveness). Multisensory displays in several taxa, such as monkeys[33], birds[34–36], fish[37], and insects[38,39] were found to be consistent with this hypothesis.

In contrast, the refinement hypothesis posits that the presence of one signal component may provide the context in which a receiver can interpret and respond to the second, with the combinations serving to disambiguate meanings (i.e. functions) when these partly overlap[17,29]. For instance, adding a signal (e.g. facial expression) to another one (e.g. gesture) may affect the likelihood of a certain interaction outcome (e.g. affiliative behaviour)[40], but also overall effectiveness (despite the fact that information of the constituent parts is non-redundant). Some evidence corroborating this hypothesis was gathered from great apes[12–14,40]. An important shortcoming of previous work, however, was that researchers did not disentangle production and perception of communicative acts, i.e. whether constituent parts varied with regard to articulators (signal production organs) or sensory channels (modalities). Teasing these apart will allow us to gain more insight into the function of multisensory signals and signal combinations in great apes.

The aim of this study was to disentangle multisensory and multicomponent communication in the great ape genus that is one of the most suitable for this avenue of research: orang-utans (*Pongo* spp.). First, the orang-utan populations of Borneo (*Pongo pygmaeus wurmbii*) and Northwest-Sumatra (i.e. Suaq and Ketambe, *Pongo abelii*) differ considerably in sociability ([41], cf. [42]) and social tolerance (Bornean orang-utans become more stressed in group settings than Sumatrans[43]). The consistently higher level of sociability in Sumatrans may lead to a greater need to refine messages conveyed in signals, and thus to more multicomponent use of communicative acts. Second, in contrast to natural environments, captive orang-utans are always in close proximity and more on the ground[9,17,29], and the lack of visual obstruction by vegetation may reduce the need for multisensory signals. Their sociability is also not constrained by food availability[44]. In the wild, individuals may have fewer opportunities to interact, and communication is hampered by arboreality and obscuring vegetation, whereas captivity enables frequent interactions and short-distance communication with conspecifics other than the mother. Third, the pairing of social partners (interaction dyad) also affects features of social interactions regardless of captive-wild and Bornean-Sumatran contrasts, e.g. due to differences in social tolerance and familiarity[45,46]. Although mothers are the most important communication partner of infant orang-utans[10,47,48], temporary associations during feeding or travelling occur, particularly if food is abundant[49,50], thus providing opportunities for social interactions beyond the mother–infant unit[51–53]. We expect that the reduced social tolerance of these dyads, and thus the lower predictability of interaction outcomes, would lead them to use more multicomponent signals.

We examined close-range communicative interactions of Bornean and Sumatran orang-utans in two wild populations and five zoos. While focal units in this study consisted of mothers and their dependent offspring, we also examined interactions with and among other members of the group/temporary association. By examining species differences related to differential sociability on one hand, and recipient-dependent factors on the other, we aimed to evaluate two major hypotheses explaining the function of multisensory and multicomponent communication (i.e. in the same sensory modality) discussed for great apes: redundancy and refinement. Since there are virtually no studies applying a similar comparative approach to any primate species, our predictions are largely exploratory.

If multisensory communicative acts indeed function as backup signals (constituent parts convey the same information as suggested by the redundancy hypothesis[9,28]), two predictions follow. First, these acts (e.g. comprising visual plus auditory acts produced in one articulator) should be more effective (i.e. more likely to result in the apparently satisfactory outcome[11], see 'Methods' section) than the single (e.g. purely visual) constituent parts, but have little or no effect on the type of outcome (i.e. dominant versus non-dominant interaction outcome, referring to whether or not the presumed goal of a particular communicative act aligned with its most common outcome, see Table 1). Second, multisensory acts should be more common in the wild than in captive settings, where semi-solitariness limits interaction opportunities and visual communication is impeded by poor visibility[17,29,30].

We now turn to multicomponent acts. If they primarily serve to refine messages, we predict that they would be used more often for non-dominant communicative goals (i.e. reducing ambiguity). For instance, if a certain communicative act is most frequently (>50%) produced towards a single interaction outcome (e.g. soliciting food transfers), but occasionally also in other contexts (e.g. initiating grooming or co-locomotion), we predict that this communicative act is accompanied by other constituent parts (e.g. specific facial expression such as a pout face, or gaze directed at recipient) more often for outcomes that are less common for that communicative act (i.e. non-dominant; in our example grooming or co-locomotion) to reduce ambiguity in these situations. Second, we predict that multicomponent acts would be more common in settings and interactions with higher uncertainty and in more varied social interactions with partners of different age-sex classes in diverse social contexts[12,14,17]. Specifically, we expect an effect of species- and dyad-dependent effect of setting: although wild individuals may use more acts associated with recipient-oriented gaze than their captive counterparts (due to lower degrees of social tolerance and thus less predictable outcomes), this effect should be more pronounced in Sumatrans (i.e. the more sociable population) and in interactions beyond the mother–offspring unit.

A secondary aim was to examine the sources of variation in the individual sensory modalities and articulators that constitute multicomponent communication in orang-utans. We predict that some modalities and articulators are more often involved in the communication process of orang-utans than others: in natural settings, dense vegetation in the canopy means that there are fewer opportunities for the direct lines-of-sight needed for visual communication, which means that we find fewer purely visual acts of facial expressions and recipient-directed gaze. As arboreal species, orang-utans are thus thought to rely less on purely visual signals than other (e.g. tactile or audio-visual) communicative means[49,54]. Moreover, previous work in wild and captive settings leads to the expectation that vocalisations play a profoundly lesser role than manual and bodily communicative acts in orang-utans close-range communication[10,48,55].

We found that that multisensory acts in orang-utans were more effective than corresponding unisensory acts and were more common in wild populations, suggesting a redundancy function. In contrast, multicomponent acts were more common for communicative acts whose presumed goal did not match with the dominant outcome, and in interactions with less socially tolerant interaction partners, requiring the usage of refining (or disambiguating) acts. Together, these findings demonstrate the

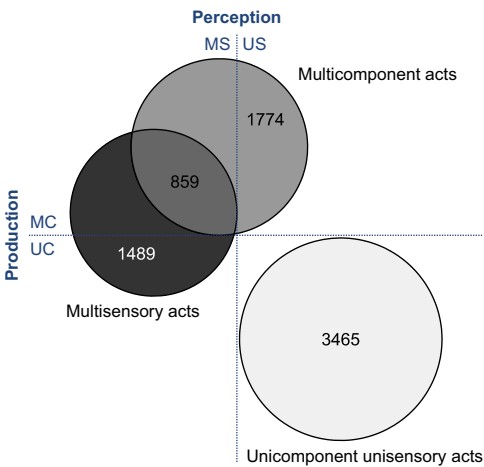

**Fig. 1 Composition of the dataset with regard to production and perception of communicative acts.** Venn diagram depicting the composition of the dataset distinguishing four exclusive categories of communicative acts: multicomponent unisensory (MC-US), multicomponent multisensory (MC-MS; overlapping area), uni-component multisensory (UC-MS), and uni-component unisensory (UC-US). Depicted values represent total sample sizes per category (Total $N = 7587$).

importance of empirically distinguishing between production and perception of communicative acts.

## Results
**Production and perception of communicative acts.** Out of the 7587 coded communicative acts, 3465 were unisensory and uni-component, 1774 multi-component but unisensory, 1489 multisensory but uni-component, and 859 both multi-component and multisensory (see Fig. 1 for Venn diagram).

Focusing on the production side first, we found that individuals used multicomponent communicative acts (i.e. acts that comprised combinations of different articulators) on average in 31% of instances, 21% of which were unisensory, and about 10 % multisensory. In terms of specific articulators, individuals used manual acts on average in 66% of observed cases, bodily acts in 24%, facial acts in 2%, vocal acts in 3% and recipient-directed gaze in 57% of cases (for detailed results in relation to species and setting, see Table 2; for sources of variation in individuals' use of specific articulators, see Supplementary Table S1 as well as Supplementary Figs. S1 and S2).

Focusing on perception, we found that individuals used multisensory communicative acts on average in 25% of observed cases, of which 15% were uni-component and about 10% were multicomponent. For specific sensory channels, we found that communicative acts contained salient visual components in 49% of cases, tactile components in 75%, auditory components in 3%, and seismic components in 1%. (for detailed results in relation to species and setting, see Table 2; for sources of variation in individuals' use of specific modalities, see Supplementary Table S1 and Supplementary Fig. S3).

**Use of multicomponent unisensory acts.** Using a GLMM with binomial error structure, we test sources of variation in the use of multicomponent acts, considering unisensory (US) acts only. Overall, the full model including the key test predictors (i.e. species x setting, kin relationship) fitted the data better than the null model (Likelihood ratio tests [LRT]: $\chi^2_5 = 127.093$, $P < 0.001$, $N = 5239$). Specifically, we found a significant interaction between species and setting (estimate ± s.e. = $-1.08 \pm 0.356$, $\chi^2_1 = 9.456$, $P = 0.002$; see Fig. 2): post hoc Sidak tests showed

that unisensory acts used by Sumatran orang-utans in either research setting were more likely to be multicomponent than those of Borneans (captivity: $-1.99 \pm 0.291$, $Z = -6.849$, $P < 0.001$, wild: $-0.91 \pm 0.215$, $Z = -4.238$, $P < 0.001$), and that unisensory acts of wild orang-utans of both species were more likely to be multicomponent than those of their captive counterparts (Borneans: $-2.36 \pm 0.314$, $Z = -7.508$, $P < 0.001$, Sumatrans: $-1.28 \pm 0.186$, $Z = -6.892$, $P < 0.001$). Unisensory communicative acts among mother–infant interactions ($-1.071 \pm 0.193$, $\chi^2_1 = 28.144$, $P < 0.001$; see Fig. 2) were less likely to be multicomponent than those among other interaction dyads. For effects of non-significant key predictors and those of control variables see Supplementary Table S2.

**Use of multisensory uni-component acts.** Next, we used an equivalent GLMM to test sources of variation in the use of multisensory acts, this time considering uni-component (UC) acts only. The full model with the key test predictors fitted the data better than the null model (LRT: $\chi^2_5 = 141.954$, $P < 0.001$, $N = 4954$). With regard to effects of specific key test predictors, we found a significant interaction between orang-utan species and research setting ($-1.306 \pm 0.391$, $\chi^2_1 = 12.041$, $P = 0.001$): post hoc Sidak tests showed that uni-component acts in wild orang-utans were more likely to be multisensory (than unisensory) compared to those of their captive counterparts regardless of species (Borneans: $-3.13 \pm 0.338$, $Z = -9.314$, $P < 0.001$, Sumatrans: $-1.82 \pm 0.216$, $Z = -8.461$, $P < 0.001$), and that captive Sumatrans produced more multisensory, uni-component acts than captive Borneans ($-1.342 \pm 0.334$, $Z = -4.021$, $P < 0.001$; see Fig. 3). For effects of other, non-significant key predictors and those of control variables see Supplementary Table S2.

**Use of multicomponent multisensory acts.** Finally, we tested sources of variation in the use of multicomponent multisensory acts, considering subsets of the dataset that consisted either only of multicomponent acts or only of multisensory acts (allowing to test the effect of the second type of integration). First, considering only the dataset of multisensory acts (i.e. contrasting MC-MS with UC-MS), we again found that the full model including the key test predictors fitted the data better than the null model (LRT: $\chi^2_5 = 45.235$, $P < 0.001$, $N = 2348$). Specifically, we found a significant interaction between species and setting ($-2.071 \pm 0.866$, $\chi^2_1 = 6.049$, $P = 0.014$; see Fig. 4): post hoc Sidak tests showed that the multisensory acts of captive Sumatran orang-utans consisted more often of multiple components than those of wild Sumatrans ($1.406 \pm 0.343$, $Z = 4.097$, $P = 0.001$) and captive Borneans ($-2.458 \pm 0.815$, $Z = -3.016$, $P = 0.003$). With regard to kinship effects, we found that multisensory acts among mother-offspring dyads were less likely to be multicomponent than those among other interaction dyads ($-0.627 \pm 0.314$, $\chi^2_1 = 3.987$, $P = 0.046$; see Fig. 4). For effects of control variables see Supplementary Table S2.

Second, considering only the dataset of multicomponent acts (i.e. contrasting MC-MS with MC-US), the full model including the key test predictors fitted the data better than the null model (LRT: $\chi^2_5 = 33.793$, $P < 0.001$, $N = 2633$). Specifically, we found a significant interaction between species and setting ($-1.916 \pm 0.726$, $\chi^2_1 = 7.664$, $P = 0.006$; see Fig. 5): post hoc Sidak tests showed that the multicomponent acts of wild Bornean orang-utans were more likely to be multisensory than those used by wild Sumatrans ($0.691 \pm 0.299$, $Z = 2.312$, $P = 0.021$) and captive Borneans ($-1.461 \pm 0.68$, $Z = -2.15$, $P = 0.032$). For the individual main effects, we found that multicomponent acts were more likely to be multisensory in interactions among mother and offspring ($0.558 \pm 0.279$, $\chi^2_1 = 3.933$, $P = 0.047$), but less likely so in interactions among maternal kin ($-0.914 \pm 0.263$, $\chi^2_1 = 11.526$, $P = 0.001$).

**Table 2 Mean percentage and SD (%) of individuals' use of communicative acts involving specific articulators and sensory modalities, and their outcomes, in relation to the research setting and orang-utan species.**

| | Captivity | | | | Wild | | | | All | |
| | Bornean | | Sumatran | | Bornean | | Sumatran | | | |
| | Mean | SD | Mean | SD | Mean | SD | Mean | SD | Mean | SD |
|---|---|---|---|---|---|---|---|---|---|---|
| Articulators | | | | | | | | | | |
| Manual | 71.6 | 10.1 | 57.6 | 13.2 | 69.5 | 17.6 | 68.1 | 20.8 | 66.4 | 17.1 |
| Bodily | 23.9 | 11.7 | 35.6 | 13.9 | 22.4 | 15.3 | 16.8 | 18.7 | 24.4 | 16.8 |
| Vocal | 0.3 | 0.8 | 2.0 | 3.7 | 5.2 | 12.9 | 2.7 | 3.8 | 2.9 | 7.6 |
| Facial | 1.5 | 2.7 | 4.6 | 4.1 | 0.6 | 1.2 | 0.8 | 1.1 | 1.8 | 3.0 |
| Gaze | 12.9 | 7.5 | 57 | 12.3 | 68.3 | 20.7 | 72.3 | 19 | 57.2 | 26.5 |
| Sensory modalities | | | | | | | | | | |
| Visual | 31.8 | 8.9 | 50.5 | 10.2 | 42.5 | 15 | 63.2 | 17.1 | 48.6 | 17.4 |
| Tactile | 72.0 | 9.2 | 66.7 | 8.4 | 92.9 | 8.4 | 63.5 | 23.0 | 74.5 | 18.6 |
| Auditory | 3.2 | 8.2 | 1.8 | 3.2 | 5.5 | 13.5 | 3.8 | 5.7 | 3.2 | 8.2 |
| Seismic | 0.0 | 0.0 | 0.4 | 0.7 | 0.6 | 1.2 | 2.3 | 4.8 | 0.9 | 2.8 |
| Communicative use | | | | | | | | | | |
| UC-US | 89.7 | 5 | 53.4 | 9.7 | 47.8 | 19.7 | 39 | 13.5 | 54.1 | 22 |
| UC-MS | 3.6 | 3.4 | 11 | 7.4 | 22.3 | 12.5 | 19.1 | 11.3 | 15.3 | 11.9 |
| MC-US | 5.9 | 4.9 | 27.3 | 9 | 16.1 | 11 | 30.3 | 11.8 | 21.1 | 13.3 |
| MC-MS | 0.7 | 1.1 | 8.2 | 4.6 | 13.8 | 13.8 | 11.7 | 7.8 | 9.5 | 9.8 |
| Outcomes | | | | | | | | | | |
| Effectiveness | 57.6 | 16.3 | 50.9 | 11.8 | 72.1 | 11.5 | 49.5 | 13.6 | 57.9 | 16.0 |
| Dominant outcome matches | 69.3 | 21.6 | 67.8 | 23.7 | 81.5 | 14.1 | 65.5 | 28.3 | 71.5 | 22.9 |

Note that communicative acts may comprise multiple articulators and sensory modalities, thus these rows do not add up to 100%.
*MC* multicomponent, *MS* multisensory, *UC* uni-component, *US* unisensory.

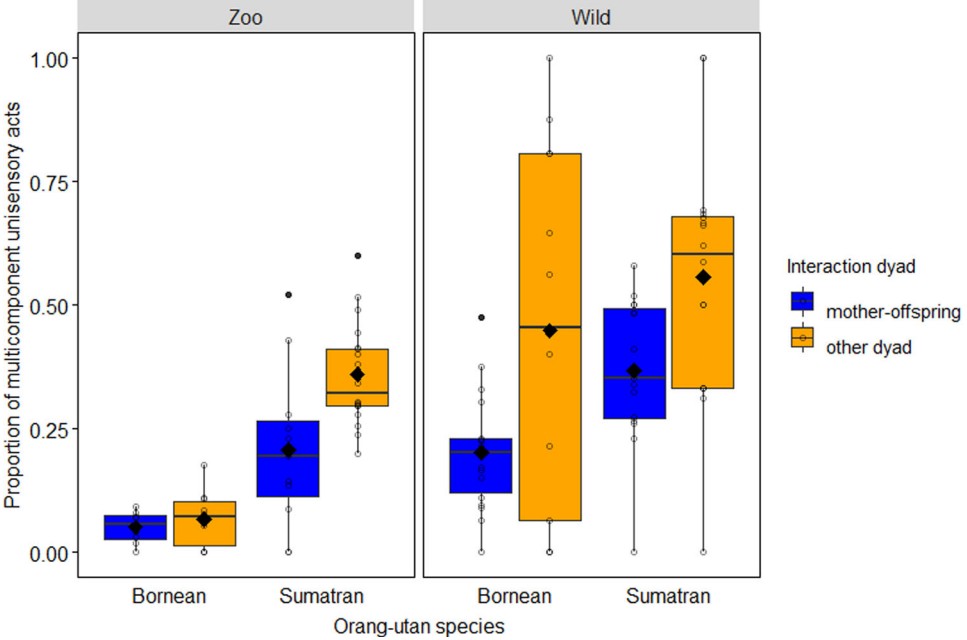

**Fig. 2 Use of multicomponent unisensory (MC-US) acts in captive versus wild orang-utans.** Proportion of multicomponent communicative acts in the sample of unisensory acts ($N = 5239$) as a function of research setting, species and interaction dyad (other dyad includes maternal kin). Proportions were significantly higher in Sumatran orang-utans compared to Borneans in both settings, and higher in wild orang-utans compared to captive ones in both species. Indicated are individual means (circles), population means (filled diamonds), medians (horizontal lines), quartiles (boxes), percentiles (2.5% and 97.5%, vertical lines) and outliers (filled dots).

**Effectiveness of multicomponent versus multisensory acts**. On average, signallers received apparently satisfactory responses to their communicative acts in 58% of observed cases (for detailed results in relation to species, setting, and communicative use see Table 2 and Supplementary Fig. S4). Using a GLMM, we tested whether the multisensory (i.e. visual plus other, tactile plus other) and multicomponent (i.e. manual plus other, bodily plus other, recipient-directed gaze plus other) use of communicative acts predicted the probability of receiving an apparently satisfactory outcome. The full models including the key test predictor fitted

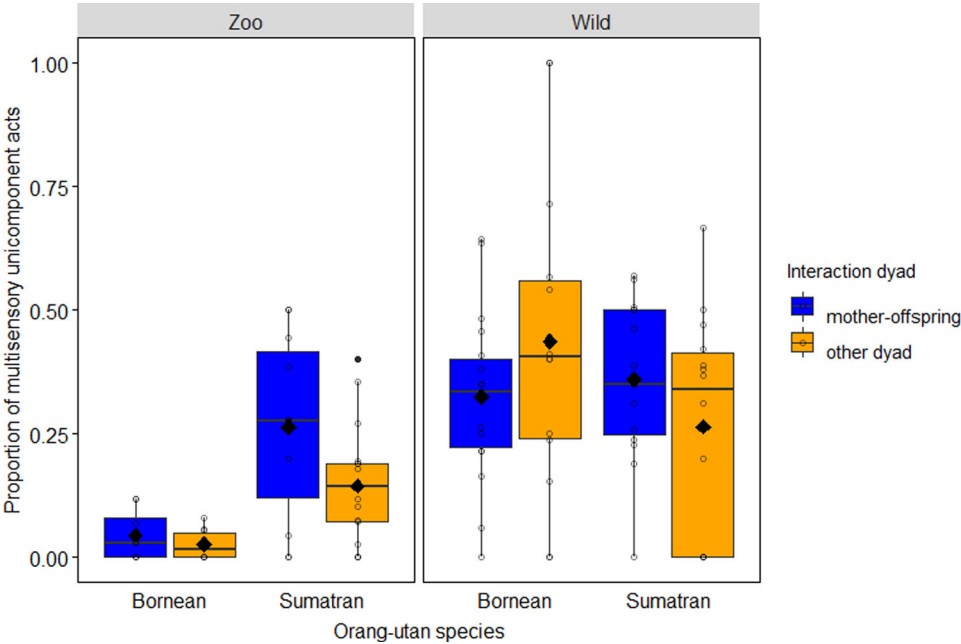

**Fig. 3 Use of multisensory uni-component (MS-UC) acts in captive versus wild orang-utans.** Proportion of multisensory communicative acts in the sample of uni-component acts ($N = 4954$) as a function of the research setting, species and interaction dyad. Proportions were significantly higher in wild orang-utans compared to captive ones regardless of species, and in captive Sumatran orang-utans compared to captive Borneans. Indicated are individual means (circles), population means (filled diamonds), medians (horizontal lines), quartiles (boxes), percentiles (2.5% and 97.5%, vertical lines) and outliers (filled dots).

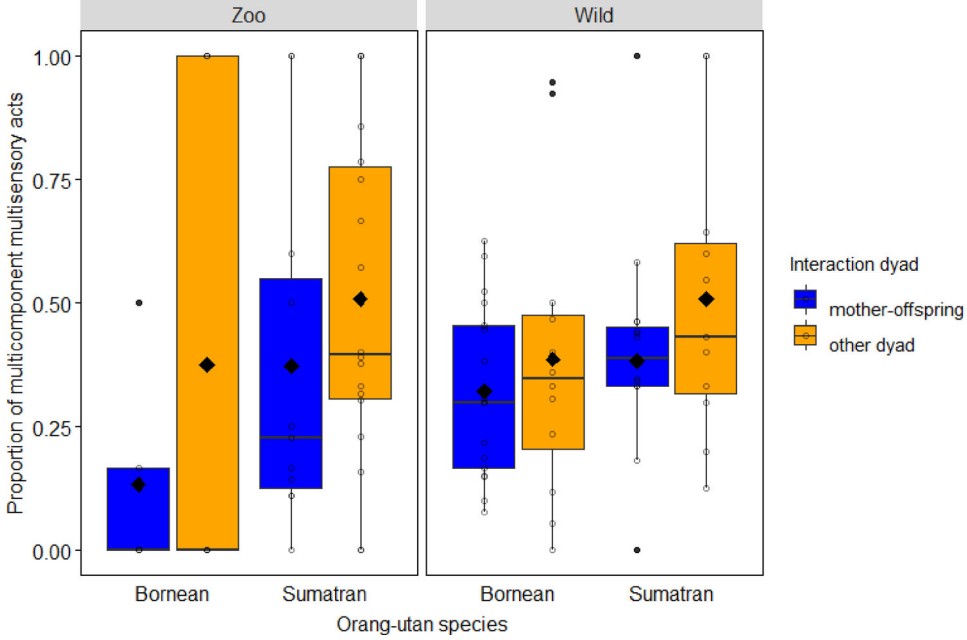

**Fig. 4 Use of multicomponent-multisensory acts (MC-MS) in captive versus wild orang-utans.** Proportion of multicomponent acts in the sample of multisensory acts ($N = 2348$) as a function of the research setting, species and interaction dyad. Proportions were significantly higher in captive Sumatran orang-utans compared to wild Sumatrans and captive Borneans. Indicated are individual means (circles), population means (filled diamonds), medians (horizontal lines), quartiles (boxes), percentiles (2.5% and 97.5%, vertical lines) and outliers (filled dots).

the data better than the null models for multisensory use of communicative acts (LRT visual plus: $\chi^2_1 = 14.458$, $P < 0.001$, $N = 2301$, see Fig. 6a; tactile plus: $\chi^2_1 = 9.692$, $P = 0.002$, $N = 3743$, Fig. 6b), as well as for multicomponent acts involving recipient-directed gaze (LRT gaze plus: $\chi^2_1 = 15.81$, $P < 0.001$, $N = 4513$, see Supplementary Fig. S5). No such effect was found for other articulators (LRT bodily: $\chi^2_1 = 0.936$, $P = 0.333$, $N = 1498$;

manual: $\chi^2_1 = 0.043$, $P = 0.837$, $N = 3037$). For effects of non-significant key predictors and those of control variables, see Supplementary Table S3. Thus, uni-component communicative acts were more likely to be effective (i.e. result in apparently satisfactory interaction outcomes) when they involved more than one sensory modality, or when recipient-directed gaze was accompanied by another articulator.

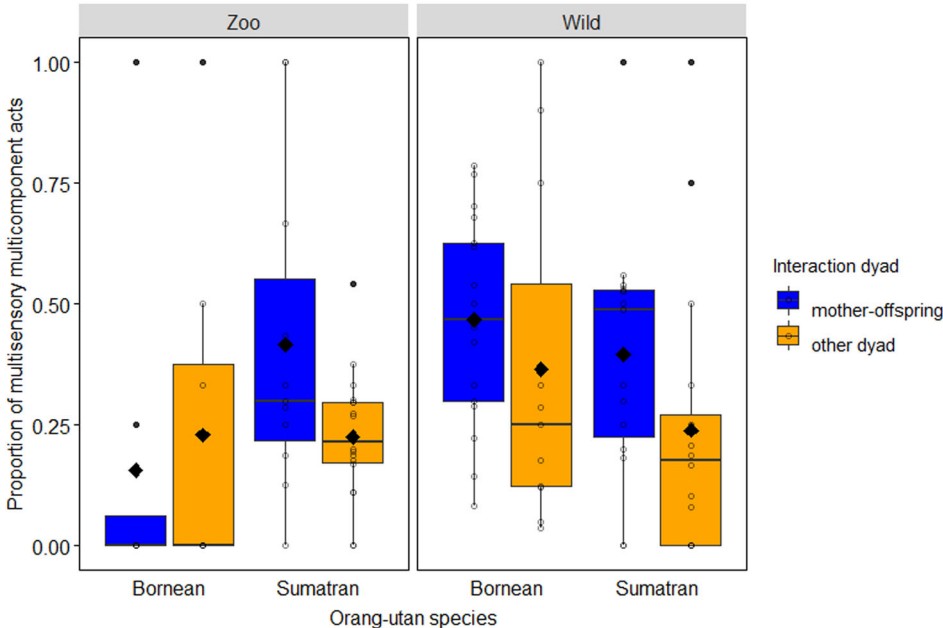

**Fig. 5 Use of multisensory multicomponent (MS-MC) acts in captive versus wild orang-utans.** Proportion of multisensory acts in the sample of multicomponent acts ($N = 2633$) as a function of the research setting, species and interaction dyad. Proportions were significantly higher in wild Bornean orang-utans compared to wild Sumatrans and captive Borneans. Indicated are individual means (circles), population means (filled diamonds), medians (horizontal lines), quartiles (boxes), percentiles (2.5% and 97.5%, vertical lines) and outliers (filled dots).

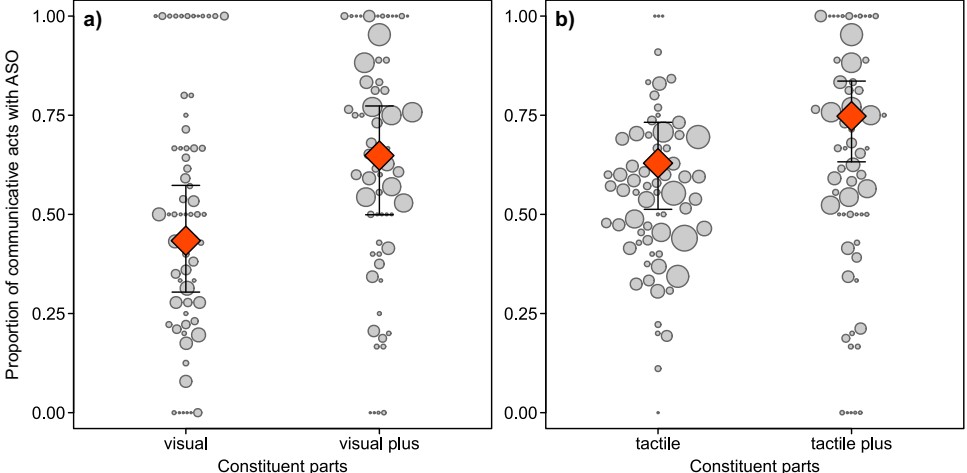

**Fig. 6 Effectiveness of multisensory uni-component (MS-UC) acts.** Proportion of uni-component communicative acts receiving an apparently satisfactory outcome (ASO) as a function of multisensory use (a: visual, $N = 2301$; b: tactile, $N = 3743$). Circles indicate individual means, with circle area representing sample size (**a** range = 1–128, **b** range = 1–175). Red diamonds depict model estimates with 95% confidence intervals (all other variables centred to a mean of zero).

**Association with dominant outcomes by multicomponent versus multisensory acts.** Communicative acts were associated with their dominant outcomes in 72% of observed cases (for detailed results in relation to species, setting, and communicative use see Table 2 and Supplementary Fig. S6). Using a GLMM, we tested whether the multicomponent (i.e. manual plus other, bodily plus other, recipient-directed gaze plus other) and multisensory (i.e. visual plus other, tactile plus other) use of communicative acts predicted whether the predominant outcome of a specific type of communicative act was matched. The key test predictor significantly enhanced the model fit for multi-component use of communicative acts except for those involving a manual component (LRT bodily plus other: $\chi^2_1 = 4.69$, $P = 0.03$, $N = 1429$, see Fig. 7a; gaze plus other: $\chi^2_1 = 6.56$, $P =$

0.01, $N = 3869$, see Fig. 7b; manual plus other: $\chi^2_1 = 0.702$, $P = 0.402$, $N = 2590$). No significant effect was found for multisensory use (LRT visual plus other: $\chi^2_1 = 3.377$, $P = 0.066$, $N = 1674$; tactile plus other: $\chi^2_1 = 3.099$, $P = 0.078$, $N = 3129$). Effects of non-significant key predictors and those of control variables are provided in Supplementary Table S4. Thus, unisensory communicative acts were significantly less likely to match dominant interaction outcomes when they involved at least two articulators (e.g. gaze plus bodily act), irrespective of setting, species or type of communicative act.

## Discussion

This study was aimed at disentangling multicomponent and multisensory communication, and at deciphering the constituting

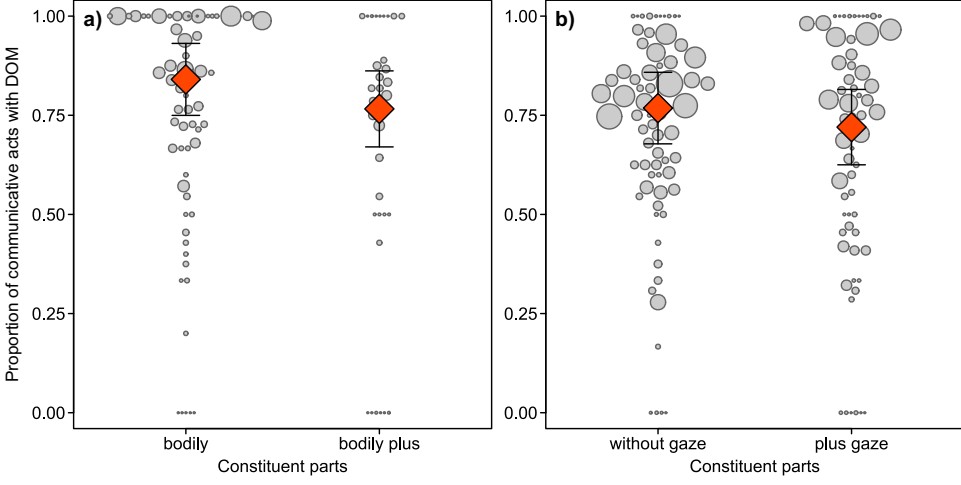

**Fig. 7 Dominant outcome matching of multicomponent unisensory (MC-US) acts.** Proportion of unisensory communicative acts whose presumed goal matched the dominant outcome (DOM) as a function of multicomponent use (**a** bodily, N = 1429; **b** recipient-directed gaze, N = 2590). Circles indicate individual means, with circle area representing sample size per individual (**a** range = 1–105, **b** range = 1–182). Red diamonds depict model estimates with 95% confidence intervals (all other variables centred to a mean of zero).

elements (that is, specific sensory modalities and articulators, respectively) in wild and captive orang-utans of two different species. More specifically, we wanted to gain insight into the functions of these two types of communicative acts by studying the effects of species and research setting on signallers' behaviour, as well as effects of multicomponent and multisensory use on responses and types of interaction outcomes.

One key finding of this study is that both multicomponent and multisensory acts differ from the respective constituent parts in both production and outcomes, and may have different functions depending on social circumstances. Thus, we can greatly improve our understanding of the function of multimodality if we tease apart the articulators and sensory channels involved.

We will first attend to our predictions and results regarding the signaller-based (articulator) perspective, and thus, multi-component communication. Our results suggest that multi-component communication may serve to reduce ambiguity, at least under certain circumstances (e.g. involving bodily acts). First and as predicted, we found that multicomponent acts (both uni- and multisensory), were more common in dyads other than mother–infant regardless of orang-utan species, but also more likely to be produced in Sumatran compared to Bornean orang-utans. The profound difference between mother-offspring and other interactions is arguably due to the high trust the signaller can have that the recipient is socially tolerant. This finding corroborates previous work on wild chimpanzees, demonstrating that purely visual, non-contact communication is more prevalent in interactions with less socially tolerant conspecifics[45,46]. As to the island difference, although orang-utans generally have fewer opportunities for social interactions than the African apes outside the mother-offspring bond (but see ref. [56] showing the overlap in solitariness between eastern chimpanzee females and North-West-Sumatran orang-utans), such social interactions are common in the populations of North-West Sumatra[41,57], and to a lesser extent in some Bornean populations for mother–infant pairs of larger matrilineal clusters[51,52]. Social interactions with conspecifics beyond the matriline are rarer in Tuanan than in Suaq[52], as are unpredictable outcomes of interactions that would require subtler communication from a larger distance. The environments that captive and wild Sumatran orang-utans inhabit, at least in this study, were also characterised by more frequent encounters with conspecifics and a thus probably a wider set of possible social partners see also[52]. Taken together,

our results strikingly demonstrate that orang-utan signallers are able to flexibly adjust their signalling to specific recipients, in line with previous work on African apes e.g. refs. [46,58].

Second, and again in line with our prediction, multicomponent unisensory acts (e.g. bodily acts accompanied by other means of the same sensory modality) were more likely to be produced when the presumed goal of the interaction did not match the dominant interaction outcome of a particular communicative act. This finding suggests that constituent parts of multicomponent acts are non-redundant and thus may serve to refine the message[17]. Human and ape communication have in common that signals are not always tightly coupled with a given referent: meaning does not only depend on the communicative act that is being used but also on the interaction history, contextual information and social aspects of the interaction[59–61]. This ambiguity facilitates the production and reuse of simple, efficient signals when contextual and social aspects of the interaction aid in inferring a specific meaning[61,62]. Importantly, by combining articulators in social interactions, interactants are able to clarify their ambiguous main signals (e.g. speech acts in humans[6,25]).

Importantly, multicomponent communication in orang-utan consisted mainly of manual/bodily acts (potential gestures sensu[63]) associated with recipient-directed eye gaze (constituting gestures according to common definitions in comparative research), rather than with vocalisations or facial expressions. Multi-component acts involving vocalisations were rare, which was probably largely due to the overall rare use of vocalisation in orang-utan close-range communication[10], but is also consistent with reports of the relatively rare use of gesture-vocalisation combinations in chimpanzees[12,13] and bonobos[14,16]. It is important to note that gaze, even though it definitely has a communicative function, often acts as a social cue rather than an intentionally produced signal. However, we do know that orang-utans are capable of controlling their gaze for bouts of intentional communication[47,48,64], suggesting that recipient-directed eye gaze serves as an important communicative articulator just as it does in humans. As an important component of social interactions, gaze can be directed at specific individuals (thereby being less ambiguous than auditory and olfactory signals), and may be used to predict another individual's behaviour[65]. We speculate that unrelated orang-utans are generally much more unpredictable in their responses, so they may have a strong tendency to visually check the emotional and attentional state of their potential interaction partners.

Turning to the recipient's (perception) perspective, in line with our expectation that the arboreal setting would impose particular communicative constraints, we found that multisensory uni-component communication was more commonly observed in wild than captive orang-utans (for Borneans, this setting contrast was also found for multisensory-multicomponent acts). Moreover, wild Bornean orang-utans more often used multisensory acts in their multicomponent communication than wild Sumatran orang-utans. Accordingly, for orang-utans the benefits of communicating in several sensory channels at once (as a backup strategy) at the expense of subtler communicative acts may be greater in the wild, where greater competition due to food scarcity may require facilitation of mutual understanding, and particularly among Bornean orang-utans.

Multisensory (uni-component) acts involving both visual and tactile components were more likely to receive apparently satisfactory responses (i.e. outcomes that resulted in the cessation of communication sensu[11,63]) than unisensory acts. At the same time, we found no evidence that multisensory acts predicted non-dominant interaction outcomes. Therefore, in orang-utans, multisensory communication seems to primarily enhance effectiveness rather than reducing ambiguity: communication through multiple sensory channels in orang-utans may facilitate detection and provide insurance that the message will be received, consistent with a redundancy function[28].

Nevertheless, our results also demonstrate that multicomponent unisensory acts (at least those involving recipient-directed gaze) can be more effective than their uni-component constituent parts. It is probably not surprising that successful disambiguation also results in more appropriate responses, which suggests that effectiveness alone is not sufficient to disentangle hypotheses for the function of multimodal communication (i.e. inferring whether signals have redundant versus non-redundant components). Nonetheless, our findings are consistent with the notion that the redundancy function applies more to multisensory signalling and thus perception features of communicative acts, whereas refinement applies more to multicomponent signalling and thus production aspects (Fig. 8). This does not mean that signals consisting of non-redundant components may not also enhance responsiveness, which is consistent with studies showing that gaze-accompanied communicative acts receive faster responses[4,5]).

We stress that these findings have to be viewed with some caution given that multisensory communication in our study mainly involved visual and tactile (rather than auditory and seismic components) in close-range interactions, and that we probably missed some low-amplitude auditory acts (e.g. vocalisations) due to environmental constraints (e.g. glass barriers in captivity or noisy surroundings in the field). However, the gestural repertoire of great apes has indeed been considered to be widely redundant[60], and studies conducted in communities of wild chimpanzees[66,67] suggest that both simultaneously and sequentially redundant signalling might play a particular role in certain developmental stages in apes, as a mechanism to learn context-appropriate communicative techniques[67].

Although previous studies on great apes mainly focused on multicomponent communication (and specifically the function of gesture-vocal combinations), not all of these communicative acts may have actually also been multisensory (e.g. audible gestures plus vocalisation when recipients are turned away or out of sight, such as drumming displays associated with pant-hoots in chimpanzees.). Captive bonobos, but not chimpanzees, have been shown to be more responsive to multicomponent (i.e. gestures combined with facial/vocal signals) than uni-component communication despite its rare usage[16]. Moreover, male bonobos use the same vocalisation (i.e. contest-hoots) in playful and aggressive contexts but add gestures to distinguish between the two[14]. For wild chimpanzees, responses to combinations of gestures and vocalisations were more likely to match the response of the gestural than the vocal components[13]. Another study showed that wild chimpanzees, after presumed goals were not achieved, switched to gesture-vocalisation combinations only if the initially single signals were vocal[12]. Moreover, a recent study on semi-wild chimpanzees' combinations of gestures and facial expressions showed that different combinations (i.e. stretched arm plus bared-teeth versus bent arm plus bared teeth) elicit different responses[40]. Thus, the evidence so far, including our own work, suggests that the combination of different articulators in great ape communication is apparently non-redundant, and serves to resolve ambiguity in the communicative act regardless of sensory modalities involved.

Multimodality seems to be functionally heterogeneous, which is also highlighted by the wealth of predictive frameworks that different behaviour researchers came up with[9,17,29,30]. If we split communicative acts by production and perception features, we get a clearer functional picture (Fig. 8): the integration of different articulators in a multicomponent act seems to primarily serve to disambiguate a message (i.e. specify meaning, as suggested by the refinement hypothesis)[12,14,17], whereas the integration of different sensory modalities in a multisensory act serves to ensure that the message arrives (i.e. enhance effectiveness, as suggested by the redundancy hypothesis)[28,33]. This is consistent with human communication, in which multisensory (audio-visual) messages were shown to be processed faster, and gestural and facial acts accompanying spoken language serve to refine and disambiguate the message conveyed in speech acts[6,25]. Given that communicative acts can be both multicomponent and multisensory, it becomes clear that both functions can be served simultaneously.

The finding that functions of multisensory and multicomponent communication may differ depending on the specific sensory modalities and articulators involved demonstrates the importance of empirically distinguishing between these forms of communication. It is therefore important that comparative studies do not compare apples with oranges: the upsurge of multimodal study designs in primate communication is timely, but comparisons with human communication will be most fruitful if the difference between production and perception features of communicative acts is explicitly addressed. Implementing such a biological meaningful comparative approach to non-human species will comprise an invaluable tool to study the origins of the human multimodal communication system.

## Methods

**Study sites and subjects**. Data were collected at two field sites and five captive facilities (zoos). We observed wild orang-utans at the long-term research sites of Suaq Balimbing (03°02′N; 97°25′E, Gunung Leuser National Park, South Aceh, Indonesia) and Tuanan (02°15′S; 114°44′E, Mawas Reserve, Central Kalimantan, Indonesia), inhabited by a population of wild Sumatran (*Pongo abelii*) and Bornean orang-utans (*Pongo pygmaeus wurmbii*), respectively. Both field sites consist mainly of peat swamp forest and show high orang-utan densities, with 7 individuals per km² at Suaq and 4 at Tuanan[68,69]. Captive Bornean orang-utans were observed at the zoos of Cologne and Munster, and at Apenheul (Apeldoorn), while Sumatran orang-utans were observed at the zoo of Zurich and at Hellabrunn (Munich; see EEP studbook for details on captive groups[70]). While captive Sumatran orang-utans were housed in groups of nine individuals each, captive Bornean groups were generally smaller and sometimes included only a mother and her offspring (e.g. Apenheul). Signallers (i.e. individuals producing communicative acts) included in this study consisted of 33 Bornean (21 wild/12 captive) and 38 Sumatran orang-utans (20 wild/18 captive). All these subjects were also recipients (i.e. individuals at which communicative acts were directed) except for one wild Sumatran subject. In addition, 11 wild Sumatran orang-utans (mostly adults) were recipients but never signallers (see Supplementary Table S5 for detailed information on subjects).

**Data collection**. Focal observations were conducted between November 2017 and October 2018 (Suaq Balimbing: November 2017–October 2018; Tuanan: January 2018–July 2018, European zoos: January 2018–June 2018). At the two field sites, they consisted of full (nest-to-nest) or partial (e.g. nest-to-lost or found-to-nest) follows of mother–infant units, whereas in zoos 6-hour focal follows were

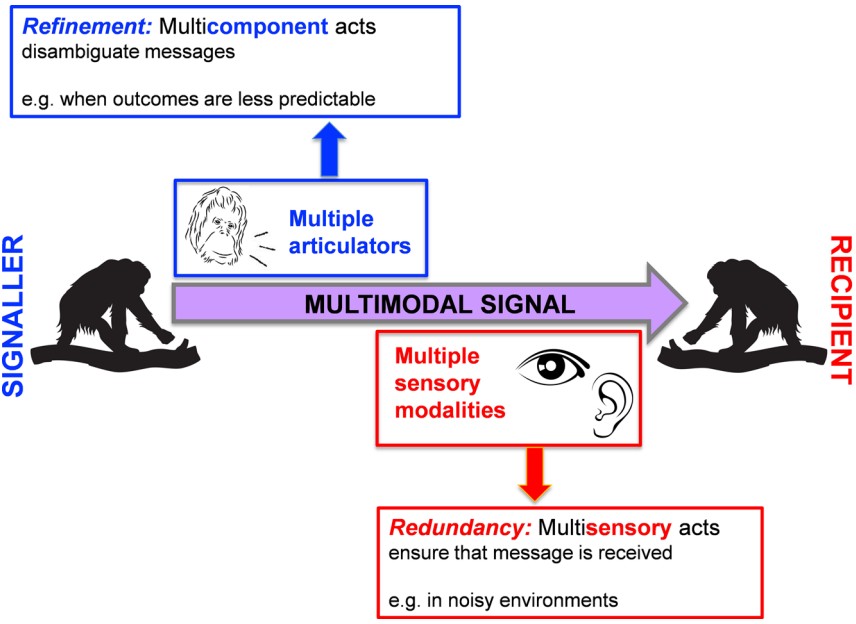

**Fig. 8 Conceptual summary of findings.** Our results suggest that refinement is the primary function of multicomponent communicative acts, whereas redundancy refers more to multisensory communicative acts.

conducted. Behavioural data were collected following an established protocol for orang-utan data collection (https://www.aim.uzh.ch/de/orangutannetwork/sfm.html), using focal scan sampling. All observers (M.F., N.B., C.F., C.W.) were trained to use this protocol and inter-observer reliability tests were conducted after each training phase. MF collected data at both field sites and two zoos ensuring the use of the same criteria during training (see ref. [71] for a recent study following the same procedure). Two different behavioural sampling methods were combined: First, intra-specific communicative interactions of all social interactions of the focal as signaller and as receiver with all partners, but also among other conspecifics present (if the focal was engaged in a non-social activity while still in full sight) were recorded using a digital High-Definition camera (Panasonic HC-VXF 999 or Canon Legria HF M41) with an external directional microphone (Sennheiser MKE600 or ME66/K6). In captive settings with glass barriers, we also used a Zoom H1 Audio recorder that was placed in background areas of the enclosure whenever possible, to enable the recording of audible communicative acts. Second, using instantaneous scan sampling at ten-minute intervals, we recorded complementary data on the activity of the focal individual, the distance and identity of all association partners, in case of social interactions the interaction partner as well as several other parameters. During ca. 1760 h of focal observations, we video-recorded more than 6300 social interactions which were subsequently screened for good enough quality to ensure video coding.

Ethical approval for our research on wild Bornean and Sumatran orang-utans was granted by the Indonesian State Ministry for Research and Technology (RISTEK, 398/SIP/FRP/E5/Dit.KI/XI/2017) and the Directorate General of Natural Resources and Ecosystem Conservation—Ministry of Environment & Forestry of Indonesia (KSDAE-KLHK, SI.70/SET/HKST/Kum.I/II/2017).

**Video coding procedure**. A total of 2655 high-quality video recordings of orang-utan interactions (wild: 1643, captive: 1012), which could each include multiple communicative acts, were coded using the programme BORIS version 7.0.4.[72] We designed a coding scheme to enable the analysis of articulators and sensory modalities involved in presumably communicative acts directed at conspecifics (i.e. close-range social behaviours that apparently served to elicit a behavioural change in the recipient and were mechanically ineffective, thus excluding practical acts such as picking up an object or acts produced with physical force[63,73]; see also Table 1) and thus included potential gestures sensu[63]. Actions that were directed at observers or achieved their presumed goal sensu[63] (apparent aim based on the individuals involved and the immediate social context) directly (e.g. nursing solicitations, infant collections) were thus excluded from the dataset. For each communicative act, we coded the following modifiers: body parts involved in production (e.g. hands or head), sensory modalities involved in perception (e.g. visual or tactile), presumed goal (e.g. share food/object, play/affiliate, co-locomotion, following the distinctions of ref. [63]), gaze direction (e.g. recipient, object), recipient's attentional state (e.g. faced towards signaller), and interaction outcome (e.g. apparently satisfactory outcome; see Supplementary Table S6 for levels and definitions of all coded variables). With regard to articulators analysed in this study (Table 1), manual communicative acts were movements executed with the limbs, bodily acts involved movements of the body, head or body postures, gaze was considered as a communicative act if it was recipient-directed or alternating

between object and recipient, facial acts involved (visible) movements of the lower face (i.e. facial expressions), and vocal acts involved the (audible) movement of vocal folds (see also ref. [8]).

To create mutually exclusive categories, we distinguished (1) uni-component unisensory acts (UC-US; one single articulator involved in the production, perceived through a single sensory modality), (2) multicomponent unisensory acts (MC-US; at least two different articulators simultaneously involved in the production, but perceived through a single sensory modality), (3) uni-component multisensory acts (UC-MS; i.e. at least two salient sensory modalities simultaneously involved in perception but produced with a single articulator), and (4) multicomponent, multisensory communicative acts (MC-MS; i.e. at least two salient sensory modalities simultaneously involved in perception and at least two articulators involved in production). We used the R package Venn Diagram[74] to visualise the proportional composition of the dataset (Fig. 1).

Adopting the terminology of Hobaiter and Byrne[11], we considered an outcome as apparently satisfactory if the signaller ceased communication and if it represented the signaller's plausible social goal. The specific types of communicative acts comprising individual and group repertoires, as well as their interaction outcomes, are reported elsewhere[75], but we used data on the dominant outcomes of communicative acts for a given research setting and orang-utan species for our test of the refinement hypothesis (Supplementary Table S7 and Supplementary Data S1).

After an initial training period of 2–4 weeks, and afterwards, in regular intervals (once a month), reliability of coding performance (using the established coding template) between at least two observers was evaluated with different sets of video recordings (10–20 clips each) using the Cohen's Kappa coefficient to ensure inter-coder reliability[76]. Trained coders (M.F., N.B., C.F., C.W., L.M., M.J.) proceeded with video coding only if at least a good level ($\kappa = 0.75$) of the agreement was found for communicative act type, articulator, sensory modality, presumed goal, and interaction outcome. For further details on the distribution of coded interactions across species, settings and interaction dyads, see Supplementary Table S8.

**Statistics and reproducibility**. For the dataset of 7587 communicative acts resulting from the coding procedure, we used Generalised Linear Mixed Models[77] with a binomial error structure and logit link function. We investigated sources of variation in four sets of response variables, referring to (a) the use of communicative acts produced with different articulators (manual, bodily, facial, vocal, recipient-directed gaze), (b) the use of communicative acts perceived via different sensory modalities (visual, tactile, auditory, seismic), (c) multicomponent and multisensory use of communicative acts, (d) effectiveness (i.e. whether or not the signaller received an apparently satisfactory response; sensu[11,63]), and (e) dominant outcome matching (i.e. whether or not the presumed goal sensu[63] of a communicative act matched the major interaction outcome [i.e. share food/object, play/affiliate, co-locomote, stop action, sexual contact, or move away] associated with it; see Table 1, and Supplementary Data S1 for the dominant outcome of communicative acts for every setting and species[75]).

In models (a), (b), and (c), we included orang-utan species (2 levels: Bornean, Sumatran), research setting (2 levels: captive, wild), and kin relationship (3 levels: mother–infant [i.e. only including unweaned immatures], maternal kin, unrelated)

as our key test predictors. Because we assumed that the effect of the research setting might depend on genetic predisposition (i.e. species), we included the interaction between these two variables in all models. To ensure valid comparisons within only one communicative perspective (i.e. production or perception) at a time, datasets included (i) only uni-component acts when testing the multisensory (MS-UC) versus unisensory (US-UC) use of uni-component acts, (ii) only unisensory acts when testing the multicomponent (MC-US) versus uni-component (UC-US) use of unisensory acts, (iii) only multicomponent acts when testing the multisensory (MS-MC) versus unisensory (US-MC) use of multicomponent acts, (iv) only multisensory acts when testing the multicomponent (MC-MS) versus uni-component (UC-MS) use of multisensory acts.

In models (d) and (e), we included multisensory (2 levels: visual/tactile only, visual/tactile plus other modality) or multicomponent use (2 levels: manual/bodily/gaze only, manual/bodily/gaze plus other articulator) as only key test predictor (communicative acts involving vocal, seismic, facial or vocal components were not common enough to allow inferential analyses; see Table 2). Analogous to the previous analyses, we considered only uni-component communicative acts when testing multisensory (MS-UC) versus unisensory (US-UC) use, and only unisensory acts when testing multicomponent (MC-US) versus uni-component (UC-US) use. We did not test effects of multicomponent-multisensory communication (MC-MS) on effectiveness and dominant outcome matches since these models did not have an acceptable stability (insufficient data for each condition).

As great ape dyadic interactions are also profoundly shaped by individual and social variables[8,12], we included further fixed effects as control predictors in the models: subjects' age class (3 levels: adult, older immature >5 years of age, young immature <5 years of age), sex (2 levels: female, male), and presumed goal (3 levels: share food/object, play/affiliate, other; as most orang-utan close-range interactions related to play or feeding; see also ref.[55]). In models (d) and (e), species, setting and kin relationship (see above) were included as control predictors rather than key test predictors as in models (a) to (c). To control for repeated measurements, the identity of the dyad, the subject and the recipient were treated as random effects. We further included group identity, video file number (accounting for the fact that communicative acts of the same interaction are non-independent), and communicative act type (e.g. touch, raise limb etc.[75]) as random effects. To keep type 1 error rates at the nominal level of 5%, we also included relevant random slope components within-subject, the recipient and/or dyad identity[78] (i.e. accounting for the non-independence of data points that pseudo-replicate slope information; depending on model stability, see below).

All models were implemented in R (v3.4.1[79]) using the function *glmer* of the package lme4[80]. To control for collinearity, we determined the Variance Inflation Factors (VIF[81]) from a model including only the fixed main effects using the function *vif* of the package car[82]. This revealed collinearity to not be an issue (maximum VIF = 2.4). To estimate model stability, we excluded the levels of random effects one at a time, ran the models again and compared the resulting estimates derived with those obtained from the respective models based on all data (see also ref.[83]). This revealed that all models were at least moderately stable, particularly for those estimates that were not close to zero (Supplementary Data S2). To test the overall significance of our key test predictors[84], we compared the full models with the respective null models comprising only the control predictors and all random effects using a likelihood ratio test[85]. To adjust for multiple comparisons, we tested interaction effects using pairwise contrasts with the function *lsmeans* (with argument 'adjust' set to 'sidak') of the package lsmeans[86]. When non-significant, these interaction terms were removed before testing the individual fixed effects. Tests of the individual fixed effects were derived using likelihood ratio tests (function *drop1* with argument 'test' set to 'Chisq').

**Reporting summary**. Further information on research design is available in the Nature Research Reporting Summary linked to this article.

## Data availability
The datasets supporting this article are available on GitHub[87], definitions of all variables and factor levels are provided in Supplementary Table S9.

## Code availability
The R code supporting this article is available on GitHub[87]. Statistical analyses were performed using R software (v3.4.1[79]) and the following packages: lme4 (v1.1-25[80]), car (v3.0-10[82]), lsmeans (v2.30-0[86]).

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

## Acknowledgements

We thank Caroline Schuppli (Suaq), Erin Vogel and Suci Utami Atmoko (Tuanan), Kerstin Bartesch (Tierpark Hellabrunn), Claudia Rudolf von Rohr (Zoo Zürich), Alexander Sliwa (Kölner Zoo), Simone Sheka (Allwetterzoo Münster) and Thomas Bionda (Apenheul Primate Park) as well as all research staff and zoo keepers for a fruitful collaboration during this study. We gratefully acknowledge Clemens Becker for providing the EEP studbook, the Indonesian State Ministry for Research and Technology (RISTEK), the Indonesian Institute of Science (LIPI), the Directorate General of Natural Resources and Ecosystem Conservation—Ministry of Environment & Forestry of Indonesia (KSDAE-KLHK), the Ministry of Internal affairs, the Nature Conservation Agency of Central Kalimantan (BKSDA), the local governments in Central Kalimantan, the Kapuas Protection Forest Management Unit (KPHL), the Bornean Orang-utan Survival Foundation (BOSF) and MAWAS in Palangkaraya. Moreover, we thank Simone Pika and Eva Luef for providing some of the technical equipment, Santhosh Totagera for coding assistance, Uli Knief for statistical advice, as well as Alexander Hausmann and Roger Mundry for providing a customised jitter-plot and model stability function, respectively. Two anonymous reviewers provided very insightful comments during the review process. MF was generously supported by the Deutsche Forschungsgemeinschaft (DFG, German Research Foundation, grant FR 3986/1-1), the Forschungskredit Postdoc (grant FK-17-106) and the A.H. Schultz Foundation of the University of Zurich, the Sponsorship Society of the German Primate Center (DPZ), the Stiftung Mensch und Tier (Freiburg) and the Christiane Nüsslein-Volhard Foundation. C.v.s. acknowledges the support of the NCCR Evolving Language Programme (SNF #51NF40_180888).

## Author contributions

M.F. and C.P.v.S. conceived of the study. M.F. designed the project, collected, coded and analysed data. N.B., C.F., C.W., L.M. and M.J. helped to collect, curate and code data. T.M.S., M.A.v.N. and C.P.v.s. provided resources. M.F. wrote the manuscript with critical inputs from M.A.v.N. and C.P.v.S. All authors approved the submission of the manuscript.

## Competing interests

The authors declare no competing interests.
