## [Peer Review File · Communications Biology]

Reviewers' comments:

Reviewer #2 (Remarks to the Author):

This is a very interesting manuscript on the complexities of social communication behavior that has the potential to interest a large number of readers. The authors are reporting on one component of a large project involving two species of orang-utans, Bornean and Sumatran, from multiple study sites (both in the wild and in captivity at zoos). They have collected an extensive data set focused on interactions within mother-infant pairs, although their data also include a wider range of subjects involved in communication interchanges. The main jist of the manuscript is that we need to analyze “multi-sensory communication acts” separately from what they refer to as “multi-articulatory communication acts”, because these two sorts of behaviors may have different ultimate functions. The data seem to support this idea, although the terminology is confusing enough that I can't easily assess the validity of the conclusion as currently presented (see comment #2 below).

I have four main comments about the manuscript overall, followed by further detailed comments and suggestions for revision.

1.) My overall impression of the work is that it is highly ambitious in scope, and because of having so many factors (species, habitat, demographics of subjects, type of communication event), there doesn't seem to be enough space within the manuscript to do justice to all of the factors. The authors do refer to another manuscript that has been submitted, with complementary data, but it is hard to assess this project without having access to those data. In particular, the actual behaviors have been left out of this manuscript (Line 239), which makes it difficult to get a full picture of the behavioral events. Behaviors here are referred to by their body part or sensory channel involved, but the actual behavior (the specific gesture or expression or vocalization, for example) is not reported. Given the impressive size and scope of this project, I realize that there are multiple papers that will be published, and it is a complex issue to figure out how to represent just part of the study without it feeling incomplete, but I think this needs some attention.

2.) My main critique and suggestion for the work is to re-think, and carefully clarify, the terminology. This may require adjusting predictions and tests. As mentioned above, this is not just a semantic issue. Because the project is focused on distinguishing between different types of communication events, the definition of those events impacts the actual outcome and our understanding of the system. The authors distinguish “multi-sensory” and “multi-articulator” acts, defined in Table 1 on page 12. They say on Line 113 that “The aim of this study was to disentangle multi-sensory and multi-articulator communication...”

My first concern is that these two terms are not mutually exclusive. The term “multi-sensory” refers to the use of multiple sensory channels (and is a commonly used term in the relevant literature, although usually without the hyphen). However, the term “multi-articulator” (not a common term) is defined by the authors in Table 1 as the use of “at least two different articulators (e.g. limbs, gaze, voice)”. By their

definition then, “multi-articulator” acts include multisensory ones (and vice-versa). It is confusing for the entire study to be set up around comparing these two groups of behaviors, when they are not mutually exclusive.

My suggestion to resolve this issue is to create mutually exclusive groups by using the terms that have been previously used in the literature: multisensory should be contrasted with unisensory, as in “unisensory, multi-component.” (Or if they have a reason to stick with “articulator”, then “unisensory, multi-articulator.”) That way you can clearly distinguish between all four logical possible cases: (1) unisensory, single component; (2) unisensory, multi-component; (3) multisensory, single component; (4) multisensory, multi-component. This would allow the authors to use their exceptionally rich data set to address two related compelling questions, with clearly non-overlapping categories: do signals with single vs multiple components (articulators) differ in usage or function?, and do signals in single vs multiple sensory channels differ in usage or function?

3.) In addition to the concern about the overlap between the two terms, I also find that the terminology varies throughout the manuscript in confusing ways. I strongly recommend that the authors clearly state their terminology at the beginning of the manuscript, and then stick to using the same terms throughout, including in all tables and figures and supplemental material. For example, the authors go back and forth between “multimodal” and “multi-sensory,” having established on line 22 that these two terms are synonyms. In that case, I would just choose one term and stick with it throughout (and use the corresponding term for the case of single components: either unimodal or unisensory).

4.) My final overall point is related to the focus of the paper: the authors make an effort to distinguish between signal production, from the point of view of the sender, and signal perception, from the point of view of the receiver. They say that the term “multi-sensory” refers to perception, and the term “multi-articulator” refers to production. While I understand this distinction, I am wondering about the logic of then considering the two categories as types of signals (here called “Communicative Acts”), and comparing between them. It feels a bit like comparing apples and oranges –which now I’m realizing the authors actually said on line 560! – if they are indeed situated at different ends of the communication dialogue (signal/signaler vs perception/receiver). At the conclusion of the manuscript (line 563), it appears that the contrast of production and perception may actually be one of the main points of their work. If that is the case, then this issue needs to be made front and center at the start of the manuscript, including the abstract, and throughout. In that case I recommend a more thorough introduction to and treatment of the topic, along the lines of what the authors have previously explained more clearly in other writings (e.g. Fröhlich and van Schaik 2018), including discussion of what other authors have written on this topic as well.

Specific Comments:

Abstract:

5) L21: at the first mention of “articulator” you should provide a definition. The definition provided (“multiplex or multi-articulator”) doesn't suffice because “multiplex” is not a common term in this

literature, and “multi-articulator” just re-states the term.

6) L29 and L30 both need referents for the comparisons (L29: “played a larger role”—larger than what? L30: “were used more” –more than what?).

Introduction:

7) The authors do a great job weaving the literature on human communication in with the literature on great apes. However I was surprised not to see other relevant primates discussed (or other non-primate animals).

8) L42-43: “recent” research is referred to, but the two citations given are from 1999 and 2000.

9) L45: citation 5 is cited for faster responses, but it appears from the reference list that citation #4 may be the one about faster responses? (If so, I wonder if these citation #s may have become swapped or off-set?)

10) L56: a particular gesture is referred to, but needs some context: describe it and mention the taxon.

11) L63-66: Here the authors attempt to clarify terminology. However, it is not clear in a number of ways. First, on L64 the reference to “both types” of communication needs specificity (what types?). And on L65-66, the term “multiplex” seems to be referenced to Higham and Hebets (2013) but that term is not in that paper. More importantly, this section comes back to the topic in my main comments (#2 and also #4) above. Are you saying that you will use “multisensory” because it is from the reception point of view, and “multi-articulator” because it is from the production point of view, or are you saying that multisensory involves multiple sensory channels but multi-articulator involves only one? If the latter, that would contradict Table 1. It would be helpful to clarify this, and to point the reader to Table 1 here or sooner; I suggest Table 1 be placed as early as possible in the text layout.

12) L71: the terminology is confusing here as well: what is being compared, when you write about comparing “the usage of uni-/multi-sensory versus uni-/multi-articulator communicative acts”? Is the comparison between the uni-versus-multi, or is it between sensory channels versus articulators? Or is it a 4-way comparison? Dealing with the terminology as suggested in my comment #2 above should help to resolve this.

13) L71 and 83: referents needed (L71: “this differentiation” – what does “this” refer to? L83: “more likely” – more likely than what?)

14) L89: The sentence starts with “For instance,” but it isn’t clear how this sentence is providing an example of the previous sentence.

15) L96: Explain this logical conclusion more. Particularly regarding why lower degrees of familiarity are involved.

16) L 102-108. Here there are two functions discussed for multi-component communication, redundancy and refinement. I was surprised to see redundancy contrasted against only one other option, though, since other authors have contrasted redundancy against a number of other options, not just refinement. Perhaps re-word this to be clear that there are other options in addition to refinement.

17) L114: I might suggest softening this to say “one of” the great ape genera most suitable, rather than claiming that Pongo is “the” most suitable genus.

18) L140: multimodal signal function, but also unimodal multi-component?

19) L144: first use of term “backup signal” should have a definition and citation.

20) L146 (&227): I am uneasy about the idea of an “apparently intended outcome”. Who decides what was apparently intended? Using what metric? More needs to be explained, either here or in the methods section, to warrant the use of this term.

21) L154: first use of term “subdominant” should have a definition and citation (particularly because the term is not being used in the usual way, relating to social dominance).

22) L156-58: explain more the rationale behind this prediction, regarding begging and non-begging interactions. This context seems rather specific to be used as a general prediction.

23) L 157 and 162: I am wondering when there would ever be a case where facial expression and gaze would not be involved? In the dark? When eyes are closed? Or do you mean a specific kind of expression and a specific kind of gaze? (Here is an example of where having the specific behaviors would be very helpful.) This comes up again later in Fig. 5 which includes a “without gaze” bar.

24) L160: explain what a “more differentiated interaction” is.

25) L167-173: Here a secondary aim is briefly mentioned. Are there predictions to go along with this aim?

Methods.

26) L176: you could call this “Study Sites and Subjects” or create a separate section for Subjects.

27) L188-191 gives details on “Signallers” and “Recipients.” Please define those terms, and give consistent summary information for both sets of subjects. Table S1 shows 71 subjects, which I assume are the signalers; Are the recipients listed somewhere too? (If it is just 10 extra individuals who were only recipients and never signalers, perhaps add them to the end of the table in a subsection, or put them in with “N Comm Acts” as zero.)

28) L203 refers the reader to “see also [ref] 44”, but it isn’t clear for what we are to see that reference.

29) L205: Focal sampling was carried out, but it also says other interactions (not involving the focal) were also collected. Can you clarify how this was done (how the focus was both on the focal animal and on the others)?

30) L219: I would call this section “Video Coding Procedure”

31) L225: might be helpful to explain “mechanically ineffective” and cite it.

32) L243-44: Give more detail: how long was the observer training period? How often were the regular intervals at which they were tested? Were all observers compared to one standard? Also please explain what “alternating datasets” means.

33) Table 1: Definitions. Communicative Act: must it necessarily be close-range? What then would a long-range signal be called? Is a behavioral change required?^[SEP]Articulator and Sensory Channel: both definitions use “such as”, which implies there are more options. I think it would be better to have a complete list, if possible.^[SEP]

34) L254: looks like a typo in the N: Table S3 shows 7587 communicative acts. I would also strike the word “around”, since this is a specific number.

35) L257-258: should these lists of modalities and articulators match what is in Table S2? Or match the R script variables in Table S7? They don’t seem to match either one. Are these lists meant to be exhaustive? In the modality list, “multi-sensory” is listed as one of the sensory modalities. Be sure this is consistent with your previous use of the term.

36) L262: the dominant outcome analysis seems to be the subject of another manuscript, and although we are pointed to the raw data here, it would be a lot easier to follow if there was a table included that clearly shows the outcomes.

37) L298: Please explain more how this controls for type 1 errors. Also curious why you didn’t use AIC; it might be helpful to justify your choice of model.

Results

38) L310: I would move this subheader (which is specific to multisensory analysis) down a couple of lines, so that the results section starts out with general results. Lines 311-313, and Figure 1 & Table 2, are general results which should be presented before the multisensory section.

39) L311-312: Typo in L311: total should be 7587. In L312, I’m surprised that there are so many multisensory but uni-articulatory acts – can you give some actual examples of behaviors that fit this category?

40) L314: is says 25% of cases were multisensory. However, this doesn't seem to match with the numbers just reported and shown in Figure 1. Figure 1 shows $1489 + 859 = 2348$ multisensory acts; 2348 is 30% of 7587, not 25%.

41) L315-17: this reporting of the results is a bit confusing due to inconsistent language: 23% were "purely visual" as opposed to 75% which "contained salient tactile components" etc. Why the difference? Does this imply that all acts contained visual components? In which case the visual modality is a constant presence?

42) Figure 1: I like this figure (with the exception of the terminology, which needs to match the text, as stated earlier), and would suggest making the sizes of the circles correspond to the N's.

43) Table 2: this shows means, but it isn't clear what they are means of. Why not include raw numbers? Why don't the numbers within each column of each section add up to 100%? They don't seem to add up to 100%, either with or without the multi-component row. A brief explanation would be handy.

44) L330-366: In general, the written results of the GLMM I find very hard to follow.

45) L339 multisensory acts were significantly less likely... than what?

46) Figs 2 & 3: Can you add indications of which are significant differences? And specify in the axis or caption what total it is a proportion of.

47) L382: it would nice to see a graph for this part as well, the body and gaze uni-vs-multi-articulator acts.

48) L384-87: How do you isolate components of a uni-articulator act? There is only one component right? This comes up again on L441.

49) L400-403: This is confusing. What does the phrase "visual multisensory" or "tactile multisensory" mean? Same question on L401 with regard to articulators.

50) L412 should there also be a result here for visual?

51) Fig 5: do "bodily act" and "gaze" include the list of those variables in table S2?

Discussion

52) L469: it says here that multi-articulator communication may reduce ambiguity "in addition to boosting effectiveness." But I thought Fig 5 showed that single articulators were more effective than multiple articulators?

53) L471-472: Confusing phrasing. The end of this sentence ("also in Sumatran... irrespective of

interaction dyad”) implies that the next sentence, about a profound difference between mother-offspring vs. other interactions, should only be true of Bornean orang-utans.

54) L 499 “ambiguous but thereby efficient”: how is ambiguity efficient?

55) L500-506. The conclusion, that “the redundancy function applies more to multi-sensory signalling, whereas refinement applies more to multi-articulator signaling”, needs more explanation. It is a bit confusing that the same evidence is used to support different conclusions for articulators vs sensory channels. (L500: “multi-articulator uni-sensory acts are more effective than their uni-articulator constituent parts.” And L438-446 says something similar for sensory systems, but leads to different conclusions.)

56) L512: this is interesting that vocalizations were so rare. All data should be given first in results section, not introduced here.

57) L526-27: this sentence is confusing because it says previous studies focused on “the function of gesture-vocal combinations”, but then it says it is “unclear to what extent communicative acts were actually also multi-sensory...” –but gesture-vocal *is* multisensory.

58) L537: “...the combination of different articulators in great ape communication is apparently non-redundant...” –does this mean multiple articulators regardless of sensory modality?

59) L547 here we find out that the multi-articulator acts “consisted to a large part of manual/bodily acts accompanied by gaze...” The fact that on the last page of the manuscript text we learn what some of the actual behaviors were is an example of how it is hard to fully understand the analysis that was presented without the data that they said is “outside the scope” of this piece. Not sure how to resolve that but it would surely be nice to know more about what the behaviors actually were.

Supplemental Material

60) Table S1. The subjects, after sorting by captive/wild and then species, seem to be randomly ordered? If so, I would re-sort them by group, and then age and sex, so that we can see at a glance the age/sex structure of each group.

61) Table S2. It says the apparent aim is based partly on the “social context” – does this mean the outcome? Also it is a bit confusing whose point of view is being represented in each column. Most of the “presumed goals” seem to be goals of the signaler, except for “Move away”, which seems to be the only outcome that is independent of the signaler?

Also I am unclear on where facial expressions would be listed in table S2?

62) Table S4. I’m assuming N should be 7587. Please also explain why the N for part (f) is smaller. I’m also wondering why the chi square values are missing for Setting and Species for all sections except part (d).

Reviewer #3 (Remarks to the Author):

The authors present a study on orang-utans communication, trying to disentangle the use and function of different articulators and channels (senses) of communication. Data consists of close-range social interactions, including mother-infant interactions, but also interactions beyond mother-infant dyads. The special focus is on comparing species (Bornean and Sumatran orang-utans) and setting (wild, captive) and whether the signals are multi- or uni-component. Multi-component communication is assumed to be focused instead on the signaller, whereas multimodal (channel, senses) communication focuses on how the recipient is processing those signals.

Results showed that multi-channel and multi-articulatory events were overall more effective than the uni-component parts and more pronounced in wild populations. In comparison to multi-sensory events, which seem generally enhance effectiveness, multi-component events depend rather on social circumstances. The results underscore the importance of differentiating between the two types when studying non-human primate communication.

The manuscript is very well written and structured, which makes it easy to follow the logical arguments. In my opinion, the introduction could benefit by adding a more explicit explanation about the distinction between "reduction of ambiguity" and "boosting effectiveness"; both do not seem to be mutually exclusive. Maybe more examples can help to clarify.

Other than that, I believe this work is a valuable contribution to non-human primate communication. Since its limitations are addressed, it gives further ideas for follow-up studies and relevant future questions. I recommend publication with minor changes. In the following, I will list some minor comments.

Statistics:

For the glmm analyses, please report attempts to check for model stability, which is usually done on the level of the estimated coefficients and standard deviations by excluding the levels of the random effects one at a time (Nieuwenhuis et al., 2012). The models are very complex, and stability is an important issue to be checked.

References:

A worth-mentioning reference in the manuscript might be Oña et al. 2019 (Oña, L. S., Sandler, W., & Liebal, K. (2019). A stepping stone to compositionality in chimpanzee communication. *PeerJ*, 7, e7623.) as it is one of the few studies in non-human primate communication which focuses on the combination of gesture and facial signals (rather than the often studied gesture-vocal combinations) and their compositional structure in chimpanzees.

Other minor comments:

L 145: delete "that" ("...we first predicted that that these...")

L 153: "If, on the other hand,..." (change "one" -> "on").

L 156: "..but also in others.." -> maybe better "...but also in other contexts.."

L 186: close the parenthesis

L 202: "see also 44" should be in parenthesis

L 213: Delete space after "parameters".

L 361: This should be "multi-articulator" I assume

L 459 "..., although this difference was more pronounced for the Bornean species." -> But not significantly, right? Maybe clarify at this point.

L 460-464: This argument is not very clear. Wasn't it found that captive Sumatrans show more multi-sensory acts compared to Borneans (Results L 337)?

Responses to Referee # 2

This is a very interesting manuscript on the complexities of social communication behavior that has the potential to interest a large number of readers. The authors are reporting on one component of a large project involving two species of orang-utans, Bornean and Sumatran, from multiple study sites (both in the wild and in captivity at zoos). They have collected an extensive data set focused on interactions within mother-infant pairs, although their data also include a wider range of subjects involved in communication interchanges. The main gist of the manuscript is that we need to analyze “multi-sensory communication acts” separately from what they refer to as “multi-articulatory communication acts”, because these two sorts of behaviors may have different ultimate functions. The data seem to support this idea, although the terminology is confusing enough that I can’t easily assess the validity of the conclusion as currently presented (see comment #2 below). I have four main comments about the manuscript overall, followed by further detailed comments and suggestions for revision.

MF et al.: Thank you for your exceptionally thorough review and the positive assessment overall. You have highlighted important shortcomings, which we hope to have addressed in our revision. In our view, your feedback has profoundly improved our manuscript, which we appreciate a lot. Please find detailed responses to your concern below.

1.) My overall impression of the work is that it is highly ambitious in scope, and because of having so many factors (species, habitat, demographics of subjects, type of communication event), there doesn’t seem to be enough space within the manuscript to do justice to all of the factors. The authors do refer to another manuscript that has been submitted, with complementary data, but it is hard to assess this project without having access to those data. In particular, the actual behaviors have been left out of this manuscript (Line 239), which makes it difficult to get a full picture of the behavioral events. Behaviors here are referred to by their body part or sensory channel involved, but the actual behavior (the specific gesture or expression or vocalization, for example) is not reported. Given the impressive size and scope of this project, I realize that there are multiple papers that will be published, and it is a complex issue to figure out how to represent just part of the study without it feeling incomplete, but I think this needs some attention.

MF et al.: We fully agree that the specific signal types and their respective outcomes are needed to grasp the full picture of orang-utan communication, especially because the interaction outcomes of individual communicative acts are critical for testing the refinement hypothesis. Therefore, we now provide an overview of our communicative repertoire with the corresponding attributes (manual, bodily, facial etc.) in an additional supplementary table (Tab. S3). Nonetheless, we still think that such a detailed analyses of communicative acts and their outcomes would explode the scope of our manuscript here. Our work on communicative repertoires is now available as a pre-print (<https://doi.org/10.1101/2021.01.19.426493>) and should be published as a peer-reviewed paper soon, thus all corresponding data are freely accessible. This was now also highlighted at the relevant mention in the present manuscript (please see page 14, line 299-303: “The specific types of communicative acts comprising individual and group repertoires as well as their interaction outcomes are reported elsewhere¹⁹, and we used the data on dominant and non-dominant outcomes of a communicative act in a given research setting and orang-utan species for our test of the refinement hypothesis (see Tab. S3 and ESM_3_outcomes).”). Please also note that we now reduced model complexity (yielding almost identical results) and conducted model stability tests for all our 14 main models, following a comment made by R2. We thus believe that the results of our statistical analyses are robust.

2.) My main critique and suggestion for the work is to re-think, and carefully clarify, the terminology. This may require adjusting predictions and tests. As mentioned above, this is not just a

semantic issue. Because the project is focused on distinguishing between different types of communication events, the definition of those events impacts the actual outcome and our understanding of the system. The authors distinguish “multi-sensory” and “multi-articulator” acts, defined in Table 1 on page 12. They say on Line 113 that “The aim of this study was to disentangle multi-sensory and multi-articulator communication...” My first concern is that these two terms are not mutually exclusive. The term “multi-sensory” refers to the use of multiple sensory channels (and is a commonly used term in the relevant literature, although usually without the hyphen). However, the term “multi-articulator” (not a common term) is defined by the authors in Table 1 as the use of “at least two different articulators (e.g. limbs, gaze, voice)”. By their definition then, “multi-articulator” acts include multisensory ones (and vice-versa). It is confusing for the entire study to be set up around comparing these two groups of behaviors, when they are not mutually exclusive. My suggestion to resolve this issue is to create mutually exclusive groups by using the terms that have been previously used in the literature: multisensory should be contrasted with unisensory, as in “unisensory, multi-component.” (Or if they have a reason to stick with “articulator”, then “unisensory, multi-articulator.”) That way you can clearly distinguish between all four logical possible cases: (1) unisensory, single component; (2) unisensory, multi-component; (3) multisensory, single component; (4) multisensory, multi-component. This would allow the authors to use their exceptionally rich data set to address two related compelling questions, with clearly non-overlapping categories: do signals with single vs multiple components (articulators) differ in usage or function?, and do signals in single vs multiple sensory channels differ in usage or function?

MF et al.: Thank you for this very insightful comment. We followed your suggestion to create four mutually exclusive categories (please see page 13, lines 287-294), and re-analysed all data related to the production of multicomponent and multisensory communicative acts (please see pages 19-23, lines 416-494). Using new binomial models, we now contrasted

a) Multisensory, unicomponent (1) and unisensory, unicomponent acts (0)

b) Multicomponent, unisensory (1) and unicomponent, unisensory acts (0)

c) Multisensory, multicomponent (1) and either multisensory, unicomponent or multicomponent, unisensory acts (0)

We also clarified that the dataset used for analysis (a) included only unicomponent acts, (b) only unisensory acts, and (c) acts that were either multisensory, multicomponent, or both (please see page 15, lines 334-341). However, this new distinction did not change our analyses and findings regarding effectiveness and dominant outcome matches, since for these analyses we “considered only unicomponent communicative acts when testing the function of multisensory acts (e.g. tactile alone versus tactile plus other) and only unisensory acts when testing multicomponent production” (pages 15-16, lines 344-344).

3.) In addition to the concern about the overlap between the two terms, I also find that the terminology varies throughout the manuscript in confusing ways. I strongly recommend that the authors clearly state their terminology at the beginning of the manuscript, and then stick to using the same terms throughout, including in all tables and figures and supplemental material. For example, the authors go back and forth between “multimodal” and “multi-sensory,” having established on line 22 that these two terms are synonyms. In that case, I would just choose one term and stick with it throughout (and use the corresponding term for the case of single components: either unimodal or unisensory).

MF et al.: We agree that our MS will benefit from a more consistent terminology. To avoid confusion, we now stick to “unisensory”/ “multisensory” to refer to acts that involve one or more sensory channels, and “unicomponent”/ “multicomponent” to refer to one or more communicative features/articulators (please see e.g. page 4, lines 68-71).

4.) My final overall point is related to the focus of the paper: the authors make an effort to distinguish between signal production, from the point of view of the sender, and signal perception, from the point of view of the receiver. They say that the term “multi-sensory” refers to perception, and the term “multi-articulator” refers to production. While I understand this distinction, I am wondering about the logic of then considering the two categories as types of signals (here called “Communicative Acts”), and comparing between them. It feels a bit like comparing apples and oranges –which now I’m realizing the authors actually said on line 560! – if they are indeed situated at different ends of the communication dialogue (signal/signaler vs perception/receiver). At the conclusion of the manuscript (line 563), it appears that the contrast of production and perception may actually be one of the main points of their work. If that is the case, then this issue needs to be made front and center at the start of the manuscript, including the abstract, and throughout. In that case I recommend a more thorough introduction to and treatment of the topic, along the lines of what the authors have previously explained more clearly in other writings (e.g. Fröhlich and van Schaik 2018), including discussion of what other authors have written on this topic as well.

MF et al.: Thank you, we agree that the distinction between production and perception has to become more clear, and highlighted this twice in the abstract (page 2, line 20-21, lines 35-36), as well as in several sections of the introduction (please see page 5, lines 77-79: “The fact that close-range communicative acts may be either multicomponent or multisensory (even if many are both) highlights the importance of teasing apart production and perception aspects of communicative acts for assessing whether they serve different communicative functions. Studying the flexible production of signals is critical as some communication systems (e.g. those of primate species) often lack the one-to-one correspondence between signal and outcome^{8,17}. On the other hand, understanding the role of perception is important because the function of animal signals is predicted by receiver psychology^{20,21} and thus by the receiver’s sensory systems^{7,22}.”. We felt that we have explicitly addressed this issue in two sections of the introduction already (but maybe not explicitly enough), when explaining that the multisensory communication is about the recipient and thus perception (page 5, line 107 onwards) and the multicomponent case takes the signaller’s perspective and thus focuses on production (page 5, line 92 onwards). We now somewhat elaborated these sections to clarify this paper’s special perspective.

Specific Comments:

Abstract:

5) L21: at the first mention of “articulator” you should provide a definition. The definition provided (“multiplex or multi-articulator”) doesn’t suffice because “multiplex” is not a common term in this literature, and “multi-articulator” just re-states the term.

MF et al.: We added a brief definition of articulators as “signal production organs” and changed the terminology in our abstract (please see page 2, line 21-23: “Consequently, the functions of integrating articulators (i.e. multicomponent acts) and sensory channels (i.e. multisensory acts) remain poorly understood.”), following the reviewer’s advice in comment #3.

6) L29 and L30 both need referents for the comparisons (L29: “played a larger role”—larger than what? L30: “were used more”—more than what?).

MF et al.: The correct referents were now added to these statements (please see page 2, line 28: “compared to captive”, and page 2, line 29-30: “than the respective unicomponent parts”).

Introduction:

7) The authors do a great job weaving the literature on human communication in with the literature on great apes. However, I was surprised not to see other relevant primates discussed (or other nonprimate animals).

MF et al.: We wanted to keep the introduction as succinct as possible and thus refrained from extensively discussing studies on multimodal communication that we covered in our previous reviews (e.g. Fröhlich & van Schaik 2018 Anim Cogn, Fröhlich et al. 2019 Biol Rev). However, we now added some relevant empirical examples from non-ape studies when introducing the two hypotheses for multimodal signal function (please see page 7, lines 127-129).

8) L42-43: “recent” research is referred to, but the two citations given are from 1999 and 2000.

MF et al.: Thank you, the misleading word was deleted accordingly (please see page 3, line 44).

9) L45: citation 5 is cited for faster responses, but it appears from the reference list that citation #4 may be the one about faster responses? (If so, I wonder if these citation #s may have become swapped or off-set?)

MF et al.: Both references (4, 5) showed effects of accompanying gaze on response latencies, hence we also moved ref 4 to the end of the sentence (please see page 3, line 47).

10) L56: a particular gesture is referred to, but needs some context: describe it and mention the taxon.

MF et al.: We now added that this gesture is “observed during the initiation of mother-offspring joint travel in chimpanzees and orang-utans^{10,11}” (please see page 3, lines 59-60).

11) L63-66: Here the authors attempt to clarify terminology. However, it is not clear in a number of ways. First, on L64 the reference to “both types” of communication needs specificity (what types?). And on L65-66, the term “multiplex” seems to be referenced to Higham and Hebets (2013) but that term is not in that paper. More importantly, this section comes back to the topic in my main comments (#2 and also #4) above. Are you saying that you will use “multisensory” because it is from the reception point of view, and “multi-articulator” because it is from the production point of view, or are you saying that multisensory involves multiple sensory channels but multi-articulator involves only one? If the latter, that would contradict Table 1. It would be helpful to clarify this, and to point the reader to Table 1 here or sooner; I suggest Table 1 be placed as early as possible in the text layout.

MF et al.: Thanks for this important suggestion. We thoroughly clarified the terminology used in our paper and inserted Table 1 at an earlier point (please see page 4, lines 66-71: “The term “multimodality” has confusingly been used to refer to communicative acts that involve multiple communicative features/articulators (e.g.^{15,16}), but also multiple sensory channels (e.g.^{7,17}). Therefore, we will henceforth refer to multicomponent and multisensory acts, respectively, to explicitly discriminate between the aspects of communicative acts that reflect production and affect perception (see Tab. 1).”). Something must have gone wrong with the EndNote citations in the manuscript, because the multiplex term was derived from the Holler & Levinson 2019 paper. We are not saying that multi-articulator acts necessarily include only one sensory channel -as we show in Figure 1, we did not consider multisensory and multi-articulator mutually exclusive categories. We now followed the reviewer’s advice (comment no. 2) and carried out new analyses that only target mutually exclusive categories: 1) unicomponent, unisensory, 2) multi-component, multi-sensory, 3) unicomponent, unisensory, and 4) multicomponent, multisensory (please see our response to comment no. 2).

12) L71: the terminology is confusing here as well: what is being compared, when you write about comparing “the usage of uni-/multi-sensory versus uni-/multi-articulator communicative acts”? Is the comparison between the uni-versus-multi, or is it between sensory channels versus articulators? Or is

it a 4-way comparison? Dealing with the terminology as suggested in my comment #2 above should help to resolve this.

MF et al.: Thank you, following your advice in comment #2, we now revised this sentence thoroughly (please see page 5, lines 84-87: “However, to date no study has explicitly investigated and compared usage of communicative acts that involved multiple rather than just one articulator (reflecting production) or sensory channel (affecting perception) – or even both – in a great ape taxon ...”). We hope the focus of our paper is clearer now.

13) L71 and 83: referents needed (L71: “this differentiation” – what does “this” refer to? L83: “more likely” – more likely than what?)

MF et al.: The correct referents were now added to these statements (please see page 5, lines 92-93 “a differentiation between the production and perception aspects of communicative acts”, and page 6, lines 113-114: “compared to a unisensory signal”).

14) L89: The sentence starts with “For instance,” but it isn’t clear how this sentence is providing an example of the previous sentence.

MF et al.: We agree and thus removed these words from the beginning of the sentence (please see page 5, line 95).

15) L96: Explain this logical conclusion more. Particularly regarding why lower degrees of familiarity are involved.

MF et al.: Thank you for pointing out the lack of clarity. We now rephrased this sentence to elaborate this statements (“...this suggests they are of particular relevance when outcomes are less predictable: when social partners are less familiar and more socially distant to each other, they are less likely to have engaged in a specific communicative interaction and disambiguation of signal meaning may become necessary” please see page 6, lines 102-105).

16) L 102-108. Here there are two functions discussed for multi-component communication, redundancy and refinement. I was surprised to see redundancy contrasted against only one other option, though, since other authors have contrasted redundancy against a number of other options, not just refinement. Perhaps re-word this to be clear that there are other options in addition to refinement.

MF et al.: This sentence was rephrased accordingly, to clarify that other functions have been discussed as well (please see page 6, line 121-122: “but see e.g. ^{17,32} for further hypotheses that have been discussed in relation to complex signal function”).

17) L114: I might suggest softening this to say “one of” the great ape genera most suitable, rather than claiming that Pongo is “the” most suitable genus.

MF et al.: This was changed accordingly, and we agree that other (e.g. fission-fusion) species may be suitable as well for this study setup (please see page 7, line 144).

18) L140: multimodal signal function, but also unimodal multi-component?

MF et al.: Thank you, we clarified this sentence accordingly using consistent terminology (please see page 8, lines 169-170: “... the function of multisensory and multicomponent (i.e. in the same sensory modality...”).

19) L144: first use of term “backup signal” should have a definition and citation.

MF et al.: We added a short definition and citation (please see page 9, lines 175: “(constituent parts convey the same information as suggested by the redundancy hypothesis ^{9,30})”).

20) L146 (&227): I am uneasy about the idea of an “apparently intended outcome”. Who decides what was apparently intended? Using what metric? More needs to be explained, either here or in the methods section, to warrant the use of this term.

MF et al.: We actually refer to apparently satisfactory outcomes as introduced by Hobaiter & Byrne (2014, Curr Biol) here (now corrected, please see page 9, line 177-178), which are increasingly used to decipher signal functions (“meanings”) in observational research on gestures. We now added a more precise definition in the methods section (please see page 14, line 297-299: “Adopting the terminology of Hobaiter and Byrne ¹¹, we considered an outcome as “apparently satisfactory” if the signaller ceased communication and if it represented the signaller’s plausible social goal.”).

21) L154: first use of term “subdominant” should have a definition and citation (particularly because the term is not being used in the usual way, relating to social dominance).

MF et al.: We replaced “subdominant” with “non-dominant”, and added a definition to clarify that we here refer to communicative acts whose presumed goal did not align with the major outcome identified for a certain communicative act (please see page 9, lines 179-181: “(i.e. dominant versus non-dominant interaction outcome, referring to whether or not the presumed goal of a particular communicative act aligned with its most common outcome, see Tab. 1”).

22) L156-58: explain more the rationale behind this prediction, regarding begging and non-begging interactions. This context seems rather specific to be used as a general prediction.

MF et al.: This sentence was now rephrased in more general terms, clarifying our prediction: The relevance of resolving ambiguity via multi-component acts is higher for less frequent social goals associated with a communicative act (please see page 9, lines 188-194: “For instance, if a certain communicative act is most frequently (>50%) produced towards a single presumed goal (e.g. soliciting food transfers), but occasionally also in other contexts (e.g. initiating grooming or, co-locomotion), we predict that this communicative act is then accompanied by other constituent parts (e.g. specific facial expression such as a pout face, or gaze directed towards recipient) more often for outcomes that are less common for that communicative act (i.e. non-dominant; in our example grooming or co-locomotion) to reduce ambiguity for these less common outcomes.”).

23) L 157 and 162: I am wondering when there would ever be a case where facial expression and gaze would not be involved? In the dark? When eyes are closed? Or do you mean a specific kind of expression and a specific kind of gaze? (Here is an example of where having the specific behaviors would be very helpful.) This comes up again later in Fig. 5 which includes a “without gaze” bar.

MF et al.: Indeed, we mean specific communicative acts, such as particular facial expressions and gaze direction. We now reworded the sentences and added examples accordingly (please see page 9, lines 191-192: “... accompanied by other constituent parts (e.g. specific facial expression such as a pout face, or gaze directed towards recipient))” and revised in the caption of Figure 5 (now Fig. 7) to clarify that we refer specifically to recipient-directed gaze

24) L160: explain what a “more differentiated interaction” is.

MF et al.: This sentence was rephrased accordingly, now clarifying that we mean “more varied social interactions with partners of different age-sex classes in diverse social contexts” (please see page 9, lines 195-196).

25) L167-173: Here a secondary aim is briefly mentioned. Are there predictions to go along with this aim?

MF et al.: We now rephrased this section to formulate more precise predictions for the use of specific articulators and sensory modalities in orang-utan communication (please see page 10, lines 204-213).

Methods.

26) L176: you could call this “Study Sites and Subjects” or create a separate section for Subjects.

MF et al.: We changed the title of this sub-section accordingly (please see page 10, line 216).

27) L188-191 gives details on “Signallers” and “Recipients.” Please define those terms, and give consistent summary information for both sets of subjects. Table S1 shows 71 subjects, which I assume are the signalers; Are the recipients listed somewhere too? (If it is just 10 extra individuals who were only recipients and never signalers, perhaps add them to the end of the table in a subsection, or put them in with “N Comm Acts” as zero.)

MF et al.: We revised this section accordingly, adding definitions and clarifying that signallers were also recipients and that there were additional subjects that were recipients but never signallers but recipient (please see page 11, lines 230-233: “All these subjects were also recipients (i.e. individuals at which communicative acts were directed) except for on wild Sumatran subject. In addition, 11 wild Sumatran orang-utans (mostly adult males) were recipients but never signallers (see Tab. S1 for detailed information on subjects and group compositions”). We also added these individuals to Table S1, following the reviewer’s suggestion.

28) L203 refers the reader to “see also [ref] 44”, but it isn’t clear for what we are to see that reference.

MF et al: We added “for a recent study following same procedure” to clarify that we adopted this training procedure in a previous study that did not focus on communication (please see page 11, line 245).

29) L205: Focal sampling was carried out, but it also says other interactions (not involving the focal) were also collected. Can you clarify how this was done (how the focus was both on the focal animal and on the others)?

MF et al: Especially in the wild, but also in captive settings, a large proportion of orang-utans’ activity budget consists of non-social activities such as feeding and resting. We now clarified in the text that we recorded interactions of other individuals in association, as long as “the focal was engaged in a non-social activity while still in full sight” (please see page 12, lines 248-249).

30) L219: I would call this section “Video Coding Procedure”

MF et al: The subtitle was changed accordingly (please see page 12, line 263).

31) L225: might be helpful to explain “mechanically ineffective” and cite it.

MF et al: Thank you, we added references and a clarifying statement (please see page 12, line 269-270: “...thus excluding practical acts such as picking up an object or those produced with physical force^{46,47}”).

32) L243-44: Give more detail: how long was the observer training period? How often were the regular intervals at which they were tested? Were all observers compared to one standard? Also please explain what “alternating datasets” means.

MF et al: This section was now thoroughly revised to provide more detail (please see page 14, lines 306-308: “After an initial training period of two to four weeks, and afterwards in regular intervals (ca. once a month), consistency of coding performance (using the established coding

template) between at least two observers was evaluated with different sets of video recordings (10 to 20 clips each) ...”

33) Table 1: Definitions. Communicative Act: must it necessarily be close-range? What then would a long-range signal be called? Is a behavioral change required? Articulator and Sensory Channel: both definitions use “such as”, which implies there are more options. I think it would be better to have a complete list, if possible.

MF et al.: To avoid any confusion, we now deleted “close-range” and the last part on behavioural changes from this definition. “Such as” was removed from the definitions, as the respective lists are exhaustive for our study (please see Tab. 1).

34) L254: looks like a typo in the N: Table S3 shows 7587 communicative acts. I would also strike the word “around”, since this is a specific number.

MF et al.: Thank you, this was corrected accordingly (please see page 14, line 315).

35) L257-258: should these lists of modalities and articulators match what is in Table S2? Or match the R script variables in Table S7? They don’t seem to match either one. Are these lists meant to be exhaustive? In the modality list, “multi-sensory” is listed as one of the sensory modalities. Be sure this is consistent with your previous use of the term.

MF et al.: Here we list all the response variables we analysed using GLMMs. We restructured this sentence slightly to clarify that “multisensory” is not supposed to be one of the sensory modalities (please see pages 14-15, lines 3187-322: “referring to (a) the use of communicative acts produced with different articulators (manual, bodily, facial, vocal, recipient-directed gaze), (b) the use of communicative acts perceived via different sensory modalities (visual, tactile, auditory, seismic), (c) multicomponent and multisensory use of communicative acts (multisensory unicomponent, multicomponent unisensory, multisensory multicomponent),”). In addition, we revised Tab. S9 (former Tab. S7) to make sure the same terms appear there.

36) L262: the dominant outcome analysis seems to be the subject of another manuscript, and although we are pointed to the raw data here, it would be a lot easier to follow if there was a table included that clearly shows the outcomes.

MF et al.: We now referred to an additional table listing all communicative acts and their dominant outcomes (Tab. S3), and listed all possible interaction outcomes here (please see page 15, lines 324-329: “[i.e. share food/object, play/affiliate, co-locomote, stop action, sexual contact, or move away]”).

37) L298: Please explain more how this controls for type 1 errors. Also curious why you didn’t use AIC; it might be helpful to justify your choice of model.

MF et al.: We now added that random slopes allow “accounting for the non-independence of data points that pseudo-replicate slope information”, thus allowing individuals to differ in the slopes of their responses (please see page 16, lines 362-363). We specified our models based on key predictors of interest and a priori knowledge about important confounding variables. Comparing full models with a reduced model (i.e. lacking the key test predictors) with a likelihood ratio test before testing the individual main effects is a common procedure in our field (Forstmeier & Schielzeth 2011, Behav Ecol) and taught by Dr. Roger Mundry at MPI for Evolutionary Anthropology. The first author has been consistently applying this procedure for many years, as it seems to produce robust and conservative results. Following a comment made by Reviewer #2, we now also conducted model stability tests which showed that the implemented models were robust.

Results

38) L310: I would move this subheader (which is specific to multisensory analysis) down a couple of lines, so that the results section starts out with general results. Lines 311-313, and Figure 1 & Table 2, are general results which should be presented before the multisensory section.

MF et al.: Thank you for catching this mistake. We have now moved the subheader to a lower line as suggested (please see page 19, line 416) and added a new subheader for the descriptive results (please see page 17, line 385).

39) L311-312: Typo in L311: total should be 7587. In L312, I'm surprised that there are so many multisensory but uni-articulatory acts – can you give some actual examples of behaviors that fit this category?

MF et al.: Thank you for catching this mistake, which we have now corrected (please see page 17, line 386). All communicative acts (e.g. manual or bodily) that are perceived through more than one sensory modality (e.g. shake object, dangle, poke with object, hit/kick, all produced in the recipient's visual field) fit in this category. In the paper, an example (the loud scratch) has been provided in the introduction (page 3, lines 59-60).

40) L314: is says 25% of cases were multisensory. However, this doesn't seem to match with the numbers just reported and shown in Figure 1. Figure 1 shows $1489 + 859 = 2348$ multisensory acts; 2348 is 30% of 7587, not 25%.

MF et al.: Please note that the Venn diagram reports absolute numbers, while the table presents individual averages. This was now clarified in the text (please see e.g. page 17, line 389), as well as the captions of Figure 1 and Table 2.

41) L315-17: this reporting of the results is a bit confusing due to inconsistent language: 23% were “purely visual” as opposed to 75% which “contained salient tactile components” etc. Why the difference? Does this imply that all acts contained visual components? In which case the visual modality is a constant presence?

MF et al.: For the sake of consistency we now report values for communicative acts that “contained visual components” as opposed to those that were “purely visual (see Tab. 2). It becomes clear from these numbers that the visual modality was not constantly present. e.g. in cases where the recipient was faced away or already in body contact with the signaller.

42) Figure 1: I like this figure (with the exception of the terminology, which needs to match the text, as stated earlier), and would suggest making the sizes of the circles correspond to the N's.

MF et al.: We changed the terminology in this figure accordingly, now referring to multisensory and multicomponent acts. Since this figure was created with a specific R package for Venn diagrams, the sample sizes do in fact correspond to N's (now clarified in the paper, please see page 13, lines 294-295).

43) Table 2: this shows means, but it isn't clear what they are means of. Why not include raw numbers? Why don't the numbers within each column of each section add up to 100%? They don't seem to add up to 100%, either with or without the multi-component row. A brief explanation would be handy.

MF et al.: These numbers represent individual means, which was now clarified in response to comment #40. We prefer to show individual means rather than raw numbers to account for the repeated measures of the same individuals here and are thus better linked to inferential analyses and figures. We now clarified that numbers will not add up to 100% since the values represent communicative acts that may contain more than one sensory channel/articulator may be involved (see caption of Tab. 2). We now added three new rows (unicomponent multisensory, multicomponent

unisensory, multicomponent multisensory) to distinguish between exclusive categories of communicative acts as suggested in comment #2.

44) L330-366: In general, the written results of the GLMM I find very hard to follow.

MF et al.: We now thoroughly rephrased these sections to better clarify the procedure we followed for each analysis, which we also explain in the methods section: first step- conducting a full-null model comparison, second step – specify interaction effect (or mention removal of interaction terms if non-significant), third step – specify the individual main effects, if applicable (please see e.g. page 20, line 417-429).

45) L339 multisensory acts were significantly less likely... than what?

MF et al.: While this specific result was now deleted following our new analyses, we now rephrased these sentence to clarify the contrast to unisensory (unicomponent) acts (please see page 19, line 423-425: “unicomponent acts of wild orang-utans of both species were more likely to be multisensory (rather than unisensory) compared to those of their captive counterparts”).

46) Figs 2 & 3: Can you add indications of which are significant differences? And specify in the axis or caption what total it is a proportion of.

MF et al.: We now specified the totals for each figure in the caption (please see Figs. 2 and 3). Since we plot the raw data here and also show effects of kinship in these figures, we prefer to keep them as they are now, but added brief descriptions of the interaction effects to the caption.

47) L382: it would nice to see a graph for this part as well, the body and gaze uni- vs multiarticulator acts.

MF et al.: Since we obtained mixed results for the effectiveness of multicomponent use of communicative acts and now included three additional figures (max allowed number of tables/figures is 10), we added the graph for gaze to the ESM rather than the main manuscript (Fig. S1). Please note that model simplification (to ensure model stability) revealed that the effect of multicomponent use of bodily acts is no longer significant.

48) L384-87: How do you isolate components of a uni-articulator act? There is only one component right? This comes up again on L441.

MF et al.: Thank you for noticing this poor description of the results. The sentences were now rephrased accordingly (“Thus, unicomponent communicative acts were more likely to be effective (i.e. result in apparently satisfactory interaction outcomes) when they involved more than one sensory modality, but also when recipient-directed gaze was accompanied by another articulator.”, page 25, lines 514-517; also page 27, line 559).

49) L400-403: This is confusing. What does the phrase “visual multisensory” or “tactile multisensory” mean? Same question on L401 with regard to articulators.

MF et al.: We now replaced these binary variables with the same terms used in the figures, clarifying that we refer to the multisensory use of communicative acts that involve a specific modality in addition to another; i.e. visual plus other, manual plus other etc. (please see pages 25-26, lines 529-530).

50) L412 should there also be a result here for visual?

MF et al.: Thank you for catching that, the non-significant result for visual plus other was now reported along with the result for tactile plus other (please see page 26, line 536: “LRT visual plus other: $\chi^2_1 = 3.377$, $P = 0.066$, $N = 1674$ ”).

51) Fig 5: do “bodily act” and “gaze” include the list of those variables in table S2?

MF et al.: Thank you for this important comment, which led us to notice that we have not provided definitions of articulators in the main text. We now provided precise definitions of the five articulators included in this study in the Video Coding Section (please see page 13, lines 280-285: “With regard to articulators analysed in this study (Tab. 1), “manual” communicative acts were movements executed with the limbs, “bodily” acts involved movements of the body, head or body postures, “gaze” was considered as a communicative act if it was recipient-directed or alternating between object and recipient, “facial” acts involved movements of the lower face (i.e. facial expressions), and “vocal” acts involved the (audible) movement of vocal folds.”).

Discussion

52) L469: it says here that multi-articulator communication may reduce ambiguity “in addition to boosting effectiveness.” But I thought Fig 5 showed that single articulators were more effective than multiple articulators?

MF et al.: Please note that Figure 5 (now Fig. 7) showed results for dominant outcome matches, not effectiveness. The results in Fig. 7 show that single articulators (i.e. bodily acts and acts without recipient-directed gaze) were more likely than multiple articulators to be used when the presumed goal matched the dominant outcome for a particular signal type. We thus argue that multiple articulators are used more for rarer outcomes (refinement).

53) L471-472: Confusing phrasing. The end of this sentence (“also in Sumatran... irrespective of interaction dyad”) implies that the next sentence, about a profound difference between mother offspring vs. other interactions, should only be true of Bornean orang-utans.

MF et al.: We rephrased the sentence accordingly, clarifying that the effect of interaction dyad holds “regardless of orang-utan species” (please see page 27, line 569).

54) L 499 “ambiguous but thereby efficient”: how is ambiguity efficient?

MF et al.: We rephrased the sentence and added a sentence to clarify under which circumstances ambiguous signals can be efficient (please see pages 28-29, lines 597-599: “This ambiguity facilitates production and allows reuse of signals when contextual and social aspects of the interaction aid in inferring a specific meaning^{79,80.}”).

55) L500-506. The conclusion, that “the redundancy function applies more to multi-sensory signalling, whereas refinement applies more to multi-articulator signaling”, needs more explanation. It is a bit confusing that the same evidence is used to support different conclusions for articulators vs sensory channels. (L500: “multi-articulator uni-sensory acts are more effective than their uniaarticulator constituent parts.” And L438-446 says something similar for sensory systems, but leads to different conclusions.)

MF et al.: We now elaborated on this statement by adding a separate section to this issue (pages 30-31, lines 640-651), clarifying that i) function may differ depending on production vs. perception of signals, and ii) refinement via multicomponent signalling can also lead to higher effectiveness (e.g. recipient-directed gaze), even if constituent parts are non-redundant (i.e. have not necessarily an identical meaning when used alone; please see page 31, line 648-651: “This does not mean, however, that signals consisting of non-redundant components may not also enhance effectiveness (as suggested by studies showing that gaze-accompanied communicative acts receive faster responses”).

56) L512: this is interesting that vocalizations were so rare. All data should be given first in results section, not introduced here.

MF et al.: We deleted this statement on results accordingly (page 29, line 606). Please note that the corresponding finding (only about 3% of individuals' communicative acts comprised vocalizations) has been provided in the results section, as part of the summary section on articulators (please see page 18, line 393), as well as in Table 2.

57) L526-27: this sentence is confusing because it says previous studies focused on “the function of gesture-vocal combinations”, but then it says it is “unclear to what extent communicative acts were actually also multi-sensory...” –but gesture-vocal **is** multisensory.

MF et al.: Strictly speaking, some gesture-vocal combinations could potentially be perceived only through the auditory channel, e.g. when the recipient is faced away or located in some distance. An example would be a drumming display associated with pant-hoots in chimpanzees. However, we agree that these instances are probably rare and not relevant to these cited studies, so we toned down this statement and provided an example (please see page 32, lines 668-71: "...not all of these communicative acts may have actually also been multisensory (e.g. audible gestures plus vocalization when recipients are turned away or out of sight, such as drumming displays associated with pant-hoots in chimpanzees.).").

58) L537: “...the combination of different articulators in great ape communication is apparently nonredundant...” –does this mean multiple articulators regardless of sensory modality?

MF et al.: Thank you, this sentence was now rephrased accordingly to enhance clarity (please page 32, lines 682-684: “Thus, evidence so far, including our own work, suggests that the combination of different articulators in great ape communication, regardless of sensory modalities involved, is apparently non-redundant, and serves to resolve ambiguity in the communicative act.”).

59) L547 here we find out that the multi-articulator acts “consisted to a large part of manual/bodily acts accompanied by gaze...” The fact that on the last page of the manuscript text we learn what some of the actual behaviors were is an example of how it is hard to fully understand the analysis that was presented without the data that they said is “outside the scope” of this piece. Not sure how to resolve that but it would surely be nice to know more about what the behaviors actually were.

MF et al.: Please note that we here refer to the category of communicative acts (e.g. manual, bodily etc.), not to the specific communicative acts that constituted the repertoire (see also response to comment #1 regarding availability of repertoire data). Findings regarding the composition of communicative acts with regard to manual, bodily, vocal etc. acts were provided in the introductory section of the results part regarding articulators (please see pages 17-18, line 389-395), as well as in Table 2. Moreover, we now provided a full list of the communicative acts included in this study and their categorisation (e.g. manual/bodily/facial/vocal) as part of the ESM (Table S3). We hope this will facilitate to understand the results.

Supplemental Material

60) Table S1. The subjects, after sorting by captive/wild and then species, seem to be randomly ordered? If so, I would re-sort them by group, and then age and sex, so that we can see at a glance the age/sex structure of each group.

MF et al.: We now reordered the subjects following the reviewer's suggestion (please see Tab. S1).

61) Table S2. It says the apparent aim is based partly on the “social context” – does this mean the outcome? Also it is a bit confusing whose point of view is being represented in each column. Most of the “presumed goals” seem to be goals of the signaler, except for “Move away”, which seems to be the only outcome that is independent of the signaler? Also I am unclear on where facial expressions would be listed in table S2?

MF et al.: With “presumed goal” we mean the signaller’s goal as determined by the observer (coder), adopting the term of Cartmill & Byrne 2010 – this was now clarified in Tab. S2. The analysis of articulators (including facial acts) based on the coding scheme was now explained in the methods section (please see page 13, lines 280-285). We also revised this supplementary table to enhance clarity (please see Tab. S2).

62) Table S4. I’m assuming N should be 7587. Please also explain why the N for part (f) is smaller. I’m also wondering why the chi square values are missing for Setting and Species for all sections except part (d).

MF et al.: Thanks, this typo was now corrected. The N for part (f – gaze direction) was the same as for the other articulators – this was also corrected (see Tab. S4). Chi Square values are missing for the main effects when interaction terms are significant (and thus included in the full model), since in these cases they have no meaningful interpretation. In part (d) the interaction term had no significant effect, so we included Chi-square values for the main effects of species and setting (please see page 17, lines 377-380 for explanation of interaction term removal).

Responses to Referee # 3

In my opinion, the introduction could benefit by adding a more explicit explanation about the distinction between "reduction of ambiguity" and "boosting effectiveness"; both do not seem to be mutually exclusive. Maybe more examples can help to clarify.

MF et al.: Thank you very much for your positive assessment and this comment – indeed, these functions are not necessarily exclusive and depend on the perspective taken – production vs perception. We now added a better explanation and more examples to clarify this distinction (please see pages 6-7, lines 122-129: “The redundant signal (hereafter referred to as ‘redundancy’) hypothesis implies that the different components convey the same information⁹, facilitating the detection and processing of a message³⁰. For example, using a conspicuous signal involving multiple modalities that contain the same information (e.g. audible and visual) makes it easier to be detected by a recipient in noisy environments and can thus increase effectiveness (i.e. responsiveness). Multisensory displays in several taxa, such as monkeys³³, birds³⁴⁻³⁶, fish³⁷, and insects^{38,39} were found to be consistent with this hypothesis. In contrast, the refinement hypothesis posits that the presence of one signal component may provide the context in which a receiver can interpret and respond to the second, with the combinations serving to disambiguate meanings (i.e. functions) when these partly overlap^{17,31}. For instance, adding a signal (e.g. facial expression) to another one (e.g. gesture) may affect the likelihood of a certain interaction outcome (e.g. affiliative behaviour)⁴⁰, but also overall effectiveness (despite the fact that information of the constituent parts is non-redundant). Most evidence in line with this hypothesis was gathered from great apes^{12-14,40}”).

Statistics: For the glmm analyses, please report attempts to check for model stability, which is usually done on the level of the estimated coefficients and standard deviations by excluding the levels of the random effects one at a time (Nieuwenhuis et al., 2012). The models are very complex, and stability is an important issue to be checked.

MF et al.: Since this procedure is computationally highly demanding and influence ME (Nieuwenhuis et al., 2012) does not allow for parallelization (i.e. using multiple cores of a machine), we checked model stability for our main models with a self-written function provided by Roger Mundry (see https://github.com/keyfm/eva/blob/master/trpm8/src/glmm_stability.r) and explained the procedure in the methods part (please see page 17, lines 370-374). The results of these checks (14 models) are now presented in an additional data file on the first author’s GitHub page (please see page 36, line 759).

References: A worth-mentioning reference in the manuscript might be Oña et al. 2019 (Oña, L. S., Sandler, W., & Liebal, K. (2019). A stepping stone to compositionality in chimpanzee communication. *PeerJ*, 7, e7623.) as it is one of the few studies in non-human primate communication which focuses on the combination of gesture and facial signals (rather than the often studied gesture-vocal combinations) and their compositional structure in chimpanzees.

MF et al.: Thank you for reminding us of this relevant paper, we now referred to it both in the introduction and discussion (please see page 32, lines 679-81: “Moreover, a recent study on semi-wild chimpanzees’ combinations of gestures and facial expressions showed that different combinations (i.e. stretched arm plus bared-teeth versus bent arm plus bared teeth) elicit different responses ⁷⁴”.

Other minor comments: L 145: delete "that" ("...we first predicted that that these...")

MF et al.: The superfluous word was deleted accordingly (please see page 9, line 176).

L 153: "If, on the other hand,..." (change "one" -> "on").

MF et al.: This was corrected accordingly (please see page 9, line 186).

L 156: "..but also in others.." -> maybe better "...but also in other contexts.."

MF et al.: This was corrected accordingly (please see page 9, line 189-190).

L 186: close the parenthesis

MF et al.: This was corrected accordingly (please see page 11, line 226).

L 202: "see also 44" should be in parenthesis

MF et al.: Parentheses were added here (please see page 11, line 245).

L 213: Delete space after "parameters".

MF et al.: The superfluous space was deleted (please see page 12, line 256).

L 361: This should be "multi-articulator" I assume

MF et al.: Thank you for catching that, this was now corrected to “multicomponent” (please see page 21, line 450).

L 459 "... although this difference was more pronounced for the Bornean species." -> But not significantly, right? Maybe clarify at this point.

MF et al.: Following from the results of our new analyses, we now clarified this statement accordingly (please see pages 29-30, lines 620-624: “... we found that multisensory unicomponent communication was more commonly observed in wild than captive orang-utans (for Borneans, this setting contrast was also found for multisensory, multicomponent acts). Moreover, wild Bornean orang-utans more often used multisensory acts in their multicomponent communication than wild Sumatran orang-utans.”).

L 460-464: This argument is not very clear. Wasn't it found that captive Sumatrans show more multi-sensory acts compared to Borneans (Results L 337)?

MF et al.: Following from the results of our new analyses using exclusive categories of communicative acts (in response to comment no. 2 made by R1), this statement was now revised (see previous comment) – wild Borneans use multisensory (multicomponent) acts more than Sumatrans, which is the finding this argument is based on (please see page 28, lines 619-628).

REVIEWERS' COMMENTS:

Reviewer #2 (Remarks to the Author):

The authors have done an admirable job revising their paper based on the first review round, and addressing each of the comments in their response. The new emphasis on production vs perception is well done, with a particularly clear explanation on lines 53-56 and a terrific new figure depicting this (Fig. 8). The addition of a new Table 1 with definitions is great, as is the re-working of the data to create four mutually-exclusive conditions. The manuscript is much improved and I have only minor suggestions at this point.

Abstract: the abstract is now framed at the start (L21) and end (L35) with the idea of production vs perception. However, the middle part of the abstract describing results (L26-33) doesn't put the findings in terms of production vs perception, so this framing (in the abstract at least) feels a little hollow. It would be good to incorporate this perspective throughout the abstract.

L22: at first mention, the term "articulators" should be defined – currently it reads "articulators (i.e. multicomponent acts)" but I don't think the authors mean to define articulators as multicomponent acts. The definition from the next page on L45 ("signal production organs") could be added to L22, for example, so that it reads "articulators (signal production organs involved in multicomponent acts)".

L74, Table 1: The table of definitions is great. I am not sure though that including the two added phrases to the definitions of the final two terms is appropriate, as they aren't so much a definition as an interpretation (or prediction?). I am referring to the parts that say "Higher effectiveness is consistent with the redundancy hypothesis" and "Association with non-dominant outcomes is consistent with the refinement hypothesis." They could perhaps be included in the text as explicit predictions instead?

L84-87: The statement that no study to date has investigated these topics in great apes appears to be contradicted by the references you cite on L119-121, which discuss the work in great apes on exactly these topics. Perhaps here you might mean to say, for example, that nobody has yet examined specifically how the function of multicomponent signals compares to that of multisensory signals?

L119: suggest to add "and multisensory" after multicomponent, since the example you give (gesture-vocal) is both.

L126: noisy environments are mentioned a few times in the revised manuscript, and it could be good to give a reference or two to work being done on multimodal communication in noise.

L151: it might be helpful to briefly explain why the need for multisensory signals is reduced when in closer proximity or on the ground.

L162: likewise, it might be helpful to explain here why multicomponent signals should be used more

with reduced familiarity.

L181: a prediction is made that “multi-component acts should be more common in the wild... where semi-solitariness limits interaction opportunities and communication is hampered by arboreality and obscuring vegetation”. I think you mean multisensory, rather than multicomponent? Since vocalizations would be helpful with obscured vision?

L244. “both study sites” I presume means the two field sites? Was this also true at the captive sites?

L334-337. Why were these constraints imposed? Explain.

Results. My main suggestion for the results is that now that you have the four mutually-exclusive conditions, it would be great to see more done with that. I wonder if there is a simple overall graph or two you could show that would directly compare the four conditions (MS, US, MC, UC). For example perhaps % of acts with ASO (or with DOM) on the Y axis and each of the 4 conditions in a bar on the X axis? I realize that you dig into the data and show pieces of these various aspects later, but a simple overall comparison of some sort I think could be powerful.

L403. Figure 1 is great. I wonder if it could be even more effective if you were to add lines delineating the four quadrants, to show how it maps on to production and perception: for example, a vertical line through the figure showing, from perception, the separation of MS (on L) and US (on R), and a horizontal line showing, for production, the separation of MC (above) and UC (below). This (or something like it) might help to keep the conditions in mind when reading the results. (I sketched this into the manuscript as I was reading, attached, and I have to say that later on when you describe results, which are complex due to all the different conditions, I found it useful to refer back to my sketch to help remember the groupings.)

Discussion.

L597. Thank you for revising this sentence, although I still find it hard to understand exactly what you mean in the revised text, when you say “ambiguity facilitates production and allows reuse of signals...” (How does ambiguity facilitate the production of a signal? And what does reuse of a signal mean?)

L620 explain or specify what “particular communicative affordances” you are referring to?

Figure 8: Terrific new diagram! I might suggest making the arrow longer, so that you can make it more obvious that you are associating multiple articulators with the signaler and multiple sensory modalities with the recipient.

L653-54: in the caption I would replace both instances of “primarily refers to” with “is the primary function of”

Supplement:

New Table S3. This is good information to include. I am trying to figure out where (for which analysis) this data was used? The reason I ask is that a few of the acts, such as kiss and look at and look back at, seem to me to be more facial acts than bodily acts. I am curious why they were classified as bodily acts, and wondering what the consequence is (in terms of which analyses this affects) of classifying them as bodily acts?

Also in the caption of the table you don't need the age category codes because age is not included here.

New Table S6. I am a bit confused as to what the difference is between c and d, "Multisensory (MC)" and "Multicomponent (MS)"? (Same question on Table S9.)

I am wondering why table S6 doesn't instead show the four conditions (MS,US,MC,UC). Explanation either here or in the text would be helpful.